# Molecular basis of bacterial DSR2 anti-phage defense and viral immune evasion

Jiafeng Huang [1,2,5] ✉, Keli Zhu[1,5], Yina Gao[2], Feng Ye[1], Zhaolong Li[2], Yao Ge[1], Songqing Liu[2], Jing Yang [3] ✉ & Ang Gao [1,4] ✉

Defense-associated sirtuin 2 (DSR2) systems are widely distributed across prokaryotic genomes, providing robust protection against phage infection. DSR2 recognizes phage tail tube proteins and induces abortive infection by depleting intracellular NAD$^+$, a process that is counteracted by another phage-encoded protein, DSR Anti Defense 1 (DSAD1). Here, we present cryo-EM structures of *Bacillus subtilis* DSR2 in its apo, Tube-bound, and DSAD1-bound states. DSR2 assembles into an elongated tetramer, with four NADase catalytic modules clustered in the center and the regulatory-sensing modules distributed at four distal corners. Interestingly, monomeric Tube protein, rather than its oligomeric states, docks at each corner of the DSR2 tetramer to form a 4:4 DSR2-Tube assembly, which is essential for DSR2 NADase activity. DSAD1 competes with Tube for binding to DSR2 by occupying an overlapping region, thereby inhibiting DSR2 immunity. Thus, our results provide important insights into the assembly, activation and inhibition of the DSR2 anti-phage defense system.

The detection of phages by the bacterial innate immune system is a fundamental aspect of anti-phage defense[1–3]. Similar to pathogen-associated molecular patterns (PAMPs) sensed by the eukaryotic innate immune systems[4], bacteria have developed diverse immune mechanisms that recognize phage-associated molecular patterns (PhAMPs) to protect themselves against bacteriophages[5–12]. Well-studied PhAMPs are phage DNA and RNA sequences that directly activate various systems, including clustered regularly interspaced short palindromic repeats (CRISPR)-Cas[13,14], restriction modification (RM)[15–17], prokaryotic Argonaute (pAgo)[18–20] and other nucleic acid-based defense systems[1,2,21,22]. Moreover, numerous phage proteins have been identified as PhAMPs that also activate bacterial immune systems[5,23]. Examples include phage capsid proteins that activate the CapRel system[6], the phage-encoded Ocr protein that activates the PARIS system[24–26], the phage portal protein and the terminase that

activate the Avs system[7,27], the phage DNA packaging protein (PacK) that activates the STK system[28], and the phage major capsid protein complexed with the host elongation factor EF-Tu that activates the Lit protease[22,29].

Recently, defense-associated sirtuin (DSR) antiphage defense systems have been found to be widely expressed in bacteria, sensing phage proteins and protecting bacteria from phage infection[2,30]. The DSR2 system from *Bacillus subtilis* interacts with the *Siphoviridae* SPR tail tube to protect bacteria from an invading phage[30]. Specifically, DSR2 functions through an abortive infection strategy that triggers NAD$^+$ depletion via the N-terminal sirtuin (SIR2) domain, caused by pattern recognition of the phage tail tube following phage infection. NAD nucleosidase (NADase) activity by SIR2 domains is a key effector in bacterial abortive infection[21,31]; others include the pAgo[32,33], Thoeris[34–37] and AVAST[7,26] systems. In addition, phages have developed

[1]Key Laboratory of Molecular Medicine and Biotherapy, Aerospace Center Hospital, School of Life Science, Beijing Institute of Technology, Beijing 100081, China. [2]Key Laboratory of Biomacromolecules (CAS), National Laboratory of Biomacromolecules, CAS Center for Excellence in Biomacromolecules, Institute of Biophysics, Chinese Academy of Sciences, Beijing 100101, China. [3]Department of Neurology, Aerospace Center Hospital, Peking University Aerospace School of Clinical Medicine, Beijing 100049, China. [4]Science and Technology Innovation Center, Shandong First Medical University & Shandong Academy of Medical Sciences, Jinan, China. [5]These authors contributed equally: Jiafeng Huang, Keli Zhu. ✉e-mail: jfhuang@ibp.ac.cn; yangjing@asch.net.cn; ang.gao@bit.edu.cn

counter-strategies to evade bacterial defense systems in their ongoing competition with the host[22,38]. *Siphoviridae* phi3T and SPbeta have been reported to encode DSAD1 (DSR anti-defense 1), which binds to DSR2 and represses its NADase activity activated by the phage tail tube[30]. However, the mechanism by which phage proteins trigger NADase activation of the DSR2 system remains unclear, as does the mechanism by which phages use DSAD1 to evade recognition by the DSR2 system. Here, we report several cryo-EM structures of apo DSR2 tetramer, DSR2-Tube and DSR2-DSAD1 complexes. By combining biochemical and mutagenesis studies, we reveal the function of the bacterial DSR2 immune system and the mechanism of phage evasion. Our work provides the structural understanding of the PhAMPs-associated NADase and also lays the foundation for further mechanistic characterization of DSR2 systems and NAD+ depletion in bacterial immunity.

## Results

### Structure of apo DSR2 tetramer

To understand the structural basis of DSR2 anti-phage defense, we expressed the full-length *Bacillus subtilis* DSR2 protein in *Escherichia coli* BL21 and determined the cryo-EM structure at 4.15 Å (Fig. 1, Supplementary Table 1, Supplementary Fig. 1, 2). This structure is referred to as the apo DSR2 tetramer to distinguish it from the complex with the Tube protein (activator, Supplementary Fig. 3) and the DSAD1 (inhibitor, Supplementary Fig. 4) bound structures reported later. In this structure, four DSR2 subunits assemble into an autoinhibited homo-tetrameric assembly, forming a ~ 480 kDa bone-shaped supramolecular complex (Fig. 1A, B). Each DSR2 protomer consists of a conserved N-terminal Sirtuin (SIR2) domain,

a C-terminal domain (CTD) and a middle domain (MD) (Fig. 1A). The SIR2 domain, which is responsible for NAD+ hydrolysis contains a Rossmann-like fold, and three additional α-helices form a small triangular structural module protruding from the Rosemann-like fold (Fig. 1A). The CTD adopts a horseshoe-shaped α-helical solenoid structure, which may play a role in interacting with other proteins and mediating the assembly of protein complexes, as observed for the solenoid domain of PI3Kα involved in the docking of p85α[39]. The middle domain connects the SIR2 and the CTD domains. Two DSR2 subunits first cross over to form an X-shaped dimer, mediated by the SIR2 and MD domains. The tetramer is then assembled by arranging two such dimers in a head-to-head orientation, facilitated by the SIR2 domain (Fig. 1B, C). To identify the key residues governing DSR2 dimerization and tetramerization, we introduced glycine or serine substitutions for key amino acids involved in these interactions. The oligomerization status of the mutants was then assessed by size exclusion chromatography (SEC). The DSR2[W143G/Y148G/Y552G/V556S/F559G/N563S] mutant with disruption at the dimeric interface behaves as a monomer, whereas the DSR2[I259S/Y260G] mutant with disruption at the tetrameric interface behaves as a dimer (Supplementary Fig. 5). As expected, in the absence of activator binding, DSR2 showed no detectable in vitro NADase activity (Fig. 2A).

Another SIR2 domain-containing protein, ThsA, is an extensively studied NADase effector that is activated by the immune second messenger cADPR produced by ThsB[1,34]. DSR2 and ThsA[35] both assemble their SIR2 domains in the same way to form a tetrameric core (Fig. 1C). The SIR2 domains show a high similarity between DSR2 and ThsA, with an RMSD (Root-Mean-Square Deviation) of 2.6 Å over 274 aligned Cα atoms when superimposed (Fig. 1D). The NAD+-binding

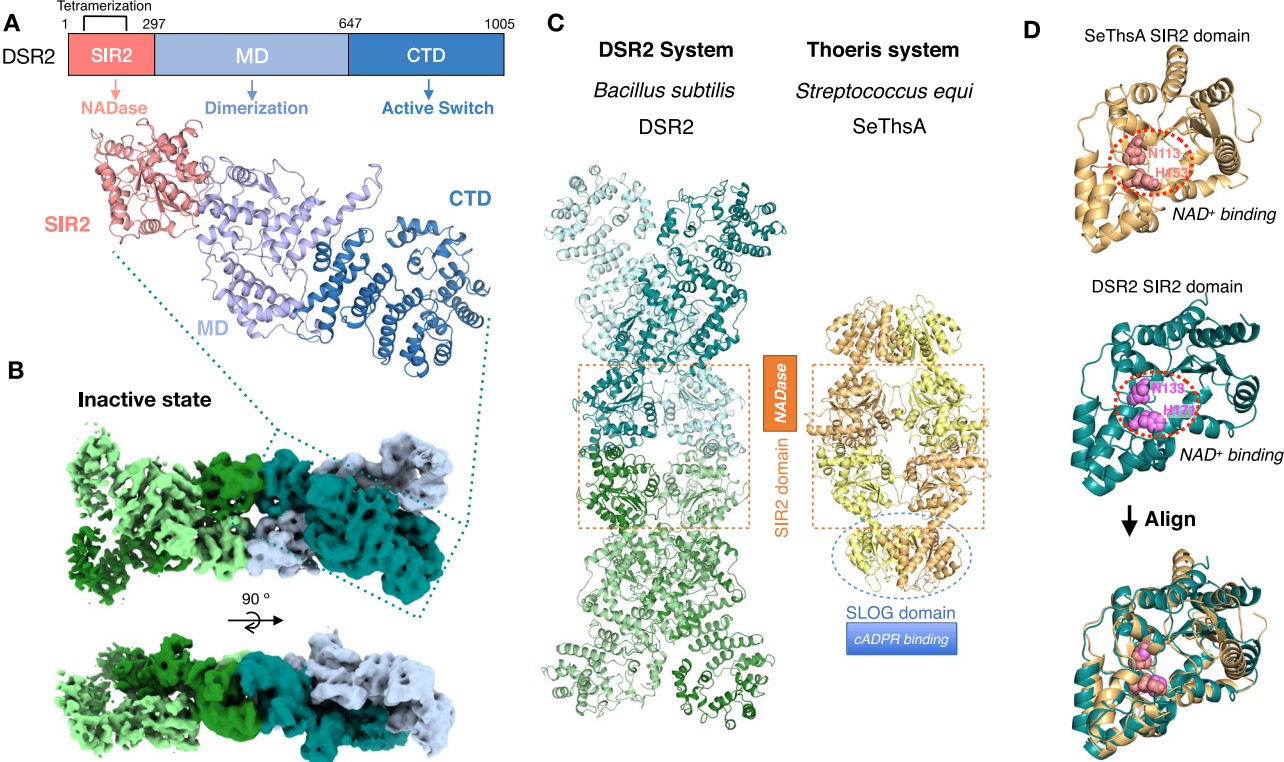

**Fig. 1 | Cryo-EM structure of the apo DSR2 tetramer. A** Domain organization of DSR2 (top) and the atomic model of DSR2 (bottom). The SIR2 domain, MD domain, and CTD domain are colored in salmon, light blue, and sky blue, respectively. **B** Top and side views of the cryo-EM density map of the apo DSR2 tetramer in the inactive state. Four DSR2s are colored in pale green, forest, light blue, and teal, respectively. The apo DSR2 tetramer has a bone-like supramolecular structure. **C** The atomic model of the apo DSR2 tetramer (left) and the SeThsA tetramer (right)

(PDB ID 7UXT). Four SIR2 domains form the tetrameric core in both the apo DSR2 tetramer and SeThsA (orange dashed boxes). The SLOG dimer of SeThsA is highlighted by a blue dashed circle. **D** Structural alignment of the SIR2 domains of apo SeThsA (light orange) and DSR2 (teal). Two types of SIR2 domains have similar structures and conserved active sites. The catalytic residues are shown as spheres, light pink for N113/H353 in SeThsA and magenta for N133/H171 in DSR2.

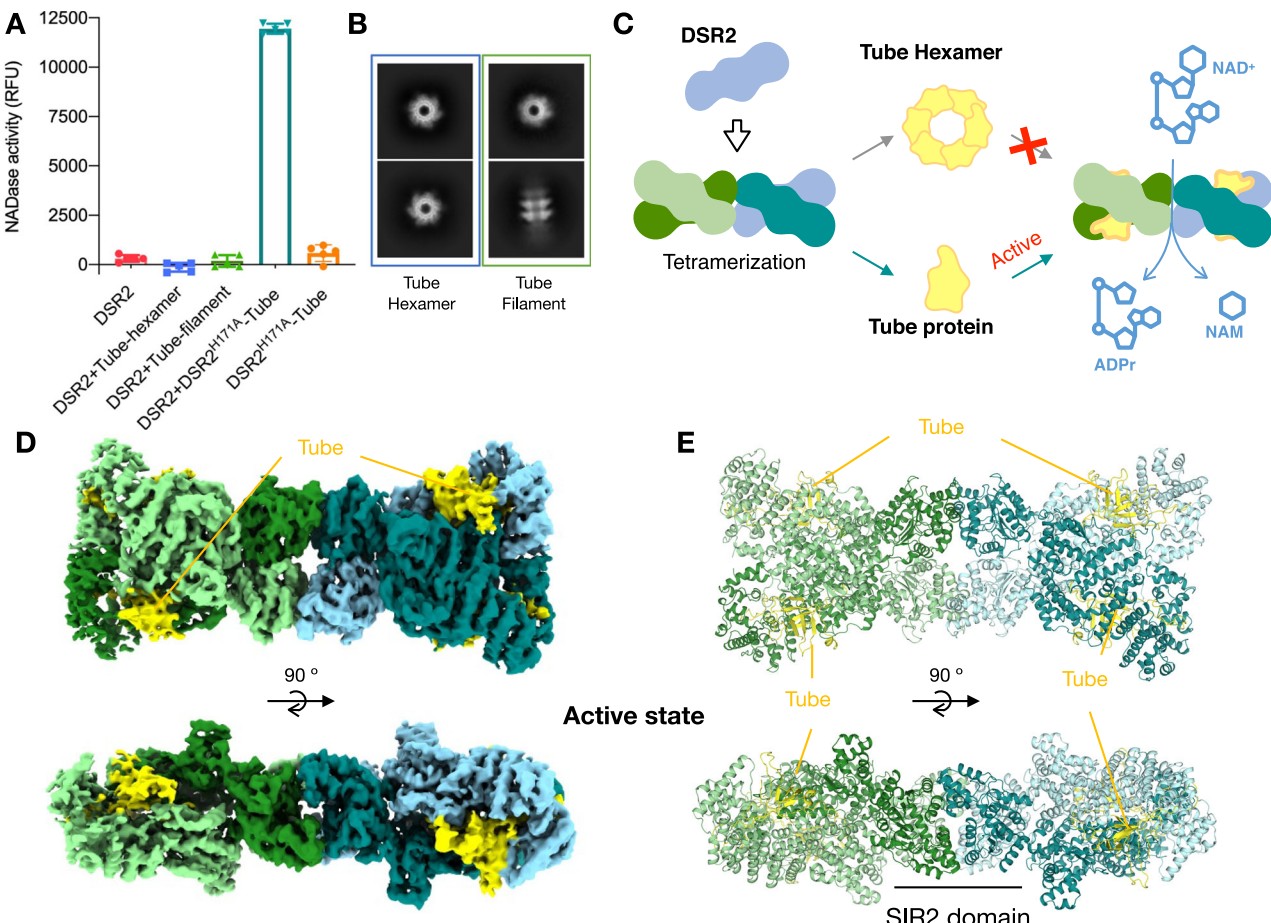

**Fig. 2 | Cryo-EM structure of DSR2^H171A-Tube. A** In vitro NAD⁺ degradation assays of DSR2 with phage Tube protein. Apo DSR2 has no detectable NAD⁺ cleavage activity. Neither the hexamer nor the filament of the Tube protein can activate the NAD⁺ activity of apo DSR2. DSR2 + DSR2^H171A-Tube complex shows NAD⁺ activity. DSR2^H171A-Tube complex abolishes NAD⁺ cleavage activity. Wild-type DSR2 was used as control. Data are presented as means ± SD ($n = 5$ independent experiments).

**B** Representative 2D class average of Tube hexamer and Tube filament. **C** Schematic diagram of DSR2 recognizing the monomeric Tube but not the hexameric Tube or filament, to trigger NAD⁺ hydrolysis. Top and side views of the cryo-EM density map (**D**) and atomic model (**E**) of DSR2^H171A-Tube complex. Four DSR2s are colored in pale green, forest, light blue, and teal; Tube proteins are colored in yellow.

sites of ThsA (N113/H153) and DSR2 (N133/H171) are conserved in both proteins. Sequence alignment reveals that residues Asn133 and His171 within the NAD⁺-binding sites of DSR2 are conserved between ThsA and other sirtuin proteins, such as pAgo, DSR1 and HerA (Supplementary Fig. 6A). Mutation of both N133A and H171A abolished the NAD⁺ cleavage activity of DSR2 even when co-expressed with Tube protein, suggesting that DSR2 and ThsA share similar NAD⁺ binding and catalytic mechanisms[30,36]. The same arrangement and similar active sites of the SIR2 domains in the apo DSR2 tetramer and inactive ThsA suggest that the apo form of DSR2 represents the inactive conformation of the DSR2 complex. Originally discovered as a NADase in abortive bacterial infections, SIR2 domains are involved in several different defense systems that protect cells from phages. The similar assembly patterns of SIR2 in the ThsA and DSR2 systems suggest that SIR2 domains in bacterial anti-phage defense systems may share similar molecular mechanisms.

**Activation of DSR2 by monomeric phage tail tube**
To obtain the DSR2-Tube complex, the apo DSR2 protein from *Bacillus subtilis* and the Tube protein from *Siphoviridae* SPR were purified separately. Unfortunately, we encountered problems in assembling the DSR2-Tube complex in vitro. The purified Tube protein does not activate the NADase activity of DSR2 (Fig. 2A). Analysis of the purified Tube protein by size exclusion chromatography (SEC) and cryo-EM shows that it exists in the form of hexamers and filaments aggregated

by hexamer units (Fig. 2B, Supplementary Fig. 7). Therefore, we hypothesized that Tube protein activates apo DSR2 in its monomeric form rather than as a hexamer or filament (Fig. 2C). To test this hypothesis, we co-expressed DSR2 and Tube protein in *E. coli* to obtain the DSR2-Tube complex. Unfortunately, expression of wild-type DSR2 with Tube protein is toxic to cells (Supplementary Fig. 8). To avoid the cytotoxicity of DSR2 NADase activation, we co-expressed a catalytic mutant DSR2^H171A lacking NADase activity with Tube protein, and successfully purified the DSR2^H171A-Tube complex (Supplementary Fig. 1C). By mixing wild-type apo DSR2 with the DSR2^H171A-Tube complex, we can clearly detect the depletion of NAD⁺ (Fig. 2A). We suspect that wild-type DSR2 may be activated either by the DSR2^H171A-Tube protomer dissociated from the tetrameric complex or by interaction with monomeric Tube released from the complex.

**Structure of DSR2-Tube supramolecular complex**
To gain a better understanding of how DSR2 recognizes the phage tail tube protein, we reconstituted the Tube-bound complex DSR2^H171A-Tube and determined its cryo-EM structure at a resolution of 3.58 Å (Fig. 2D, Supplementary Table 1, Supplementary Fig. 3). The structure of the DSR2^H171A-Tube complex shows the same "bone-shaped" architecture as the apo DSR2 tetramer and reveals a 4:4 assembly with a tetrameric core of DSR2 subunits braced on each corner by a Tube protein (Fig. 2D, E). When the hexameric Tube is superimposed on the structure of the DSR2-Tube complex, a clash between the Tube

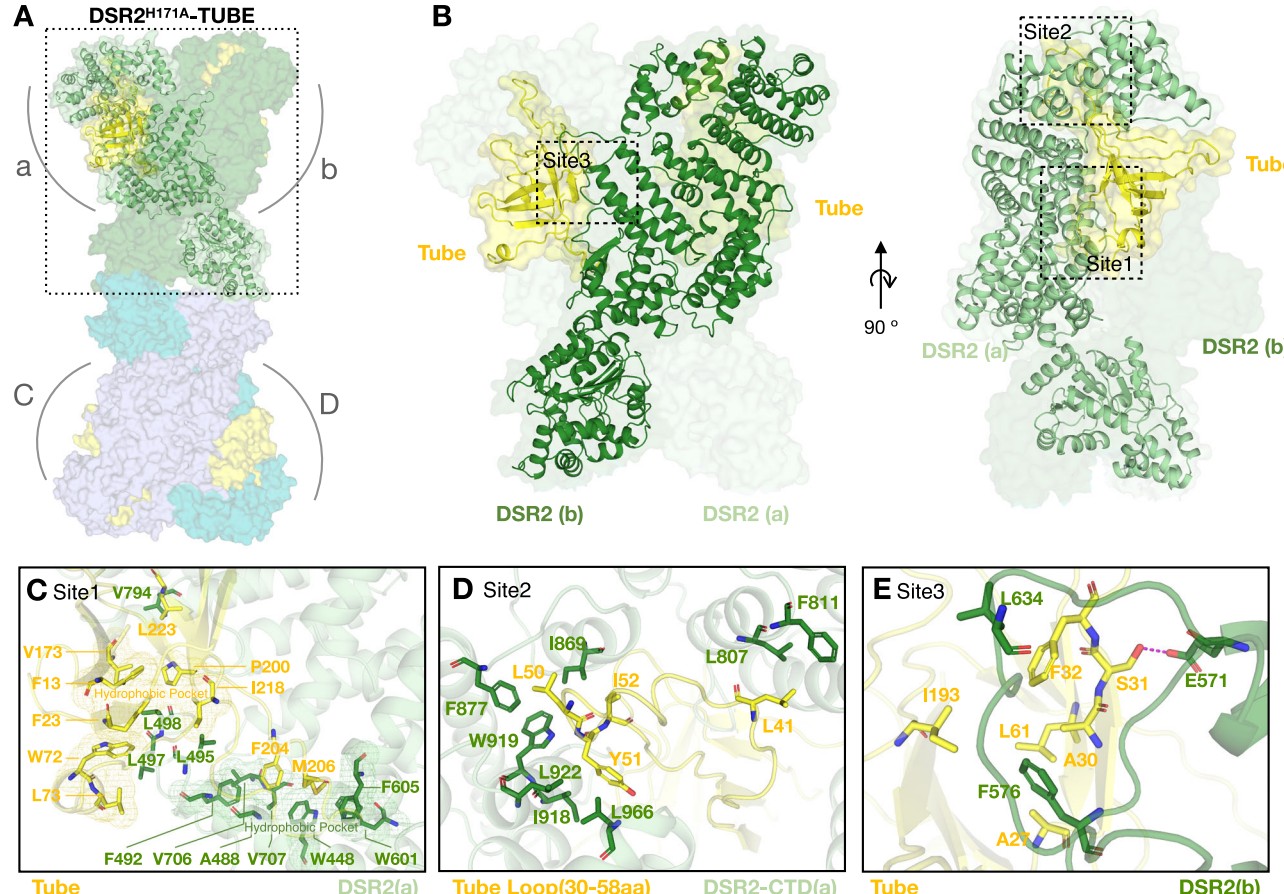

**Fig. 3 | Interaction between DSR2^H171A and phage Tube protein. A** Overall structure of the DSR2^H171A-Tube complex. Four DSR2s are colored in pale green, forest, light blue and teal; Tube proteins are colored in yellow. **B** Top and side views of the DSR2-Tube interface (zoom-in of the dashed box in panel A), same color code as in (**A**). **C–E** Detailed views of the interactions between DSR2 and Tube protein. Sites 1 to 3, (**C–E**), respectively, of hydrophobic interactions between DSR2 and Tube are showed; same color code as in (**A**). Key interacting residues are shown in stick representation.

hexamer and DSR2 is observed, explaining why only the monomeric Tube protein and not the hexameric Tube is able to activate the NADase activity of DSR2 (Supplementary Fig. 9). The phage tail tube protein is generally expressed during the later stage of the phage life cycle and aggregates into a homo-hexamer that is ready for phage assembly[23]. The apo DSR2 complex inside the cell is normally inactive, but is primed for rapid activation to combat the expression of the phage tube protein.

To understand the interactions between DSR2 and the phage tail tube protein, the cryo-EM structure of the DSR2^H171A-Tube complex was locally refined and the local structure was determined to 3.08 Å (Supplementary Fig. 10A). The DSR2^H171A-Tube complex has a large contact interface of approximately 4094 Å² × 4. Each Tube protein binds to two DSR2 proteins on the same side (3152.65 Å² for DSR2(a) and 886.14 Å² for DSR2(b), respectively) (Fig. 3A). In the Tube-DSR2 surface, the phage tube protein forms intermolecular interactions with both the middle domain and the C-terminal domain of DSR2 (Fig. 3B). The Tube-DSR2 interactions are mainly mediated by a large hydrophobic area on the middle domain. The side chains of L495/L497/L498 of DSR2(a) fits perfectly into a hydrophobic pocket of Tube (Fig. 3C). Several important residues of Tube, with bulky side chains (F13/F23/W72/L73/V173/P200/I218 of Tube), form the surface of this DSR2-binding groove and make hydrophobic contacts with DSR2(a). In addition, the loose loop region (aa 202-217) of Tube inserts into the hydrophobic groove of DSR2(a) (Fig. 3C). Specifically, the side chain of F204/M206 of Tube inserts into a hydrophobic pocket composed of four helices (α23:W448; α24:A488/F492; α30:W601/F605; α34:V706/

V707) of DSR2(a). The DSR2^L495S/L497G/L498S and Tube ^F204A/M206A mutants disrupt the interactions between DSR2 and Tube and fail to form the DSR2-Tube complex (Supplementary Fig. 11). On another binding site (Site2), the loop (aa 35-55) of Tube is completely inserted into the ring of the C-terminal circular solenoid structure DSR2, forming strong hydrophobic interactions. These interactions include L41 of Tube with L807/F811 of DSR2(a) and L50/Y51/I52 of Tube with I869/F877/I918/W919/L922/L966 of DSR2(a) (Fig. 3D). At Site3, two loop regions (aa 572-579 and aa 627-642) of DSR2(b) form hydrophobic interactions with Tube (A27/A30/S31/F32/L61/I193) (Fig. 3E). Specifically, side chains F576 and L634 of DSR2(b) insert into a hydrophobic pocket composed of Tube residues (A27, A30, F32, L61 and I193). In addition, S31 of Tube forms a strong hydrogen bond with E571 of DSR2(b). Mutations at Site2 (Tube^L50A/Y51A/I52A) and Site3 (Tube^S31A/F32A), which disrupt the interaction surfaces, prevent the formation of the DSR2-Tube complex (Supplementary Fig. 11).

## SIR2 assembly is important for DSR2 NADase function

According to size exclusion chromatography (SEC) analysis and cryo-EM structure, both the apo DSR2 and the DSR2^H171A-Tube complex form tetramers, mediated by the SIR2 tetrameric core. This suggests a critical role for the SIR2 tetramer in these complexes (Figs. 1B, 2D). In the DSR2^H171A-Tube complex, four SIR2 domains are arranged in a centrosymmetric tetragonal pattern, with each strand consisting of two SIR2 protomers stacked head-to-head (Fig. 4A). Two types of repeated interactions, side-by-side (a–b and c–d) and head-to-head (a–c and b–d) contacts, mediate the formation of the

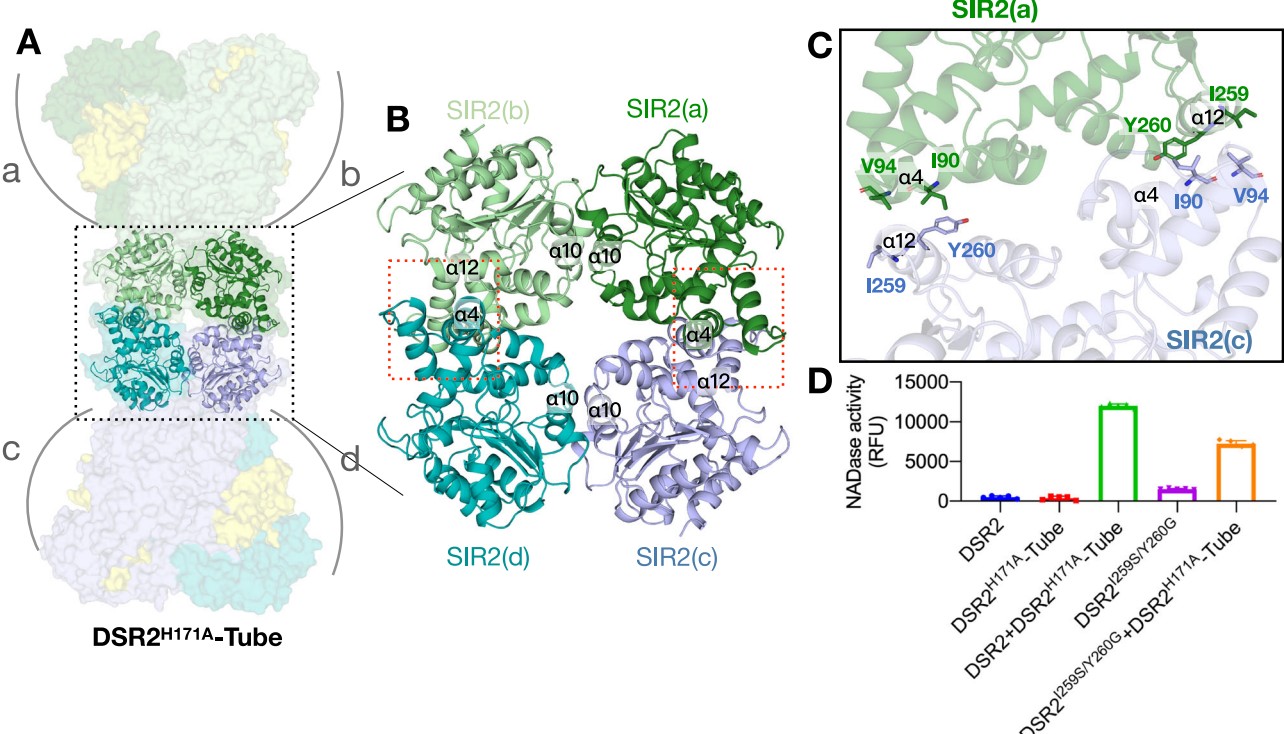

**Fig. 4 | The SIR2 assembly of the DSR2. A** Overview structure of the SIR2 assembly in the DSR2^H171A-Tube complex. **B** Tetramerization of SIR2 domains. The dashed squares show the head-to-head contacts formed by the SIR2 domains. **C** Detailed insights into the head-to-head interactions. Key interacting residues are shown in stick representation. I259 and Y260 in the SIR2 α4 helix interact with I90 and V94 in

SIR2 α12. **D** The I259S/Y260G mutation in the SIR2 assembly interface decreased NAD⁺ cleavage. Wild-type DSR2 was used as control. NADase activity was decreased in the I259S/Y260G mutant. Data are presented as means ± SD ($n = 5$ independent experiments).

SIR2 tetramer (Fig. 4B). In particular, helices α8 and α10 of SIR2 are mainly responsible for the side-by-side association (Fig. 4B). The interface of SIR2(a) and SIR2(b) is symmetrical and the inter-molecular binding is mainly contributed by hydrophobic interactions and further strengthened by polar contacts. The head-to-head contacts are primarily mediated by helices α4 and α12 (Fig. 4B). The I259 and Y260 in SIR2(a) α4 helix interact with the I90 and V94 in SIR2c α12 (Fig. 4C). Symmetrically, the I90 and V94 in SIR2(a) α4 helix also interact with the I259 and Y260 in SIR2c α12, suggesting a critical role of this hydrophobic interaction site in SIR self-association. To investigate whether the tetramerization of DSR2 is necessary for the NADase activity of the DSR2 system, an I259S/Y260G mutant was designed to disrupt the dimer-dimer interaction. As expected, DSR2^I259S/Y260G appears in a dimeric form according to the SEC analysis (Supplementary Figs. 5,12) and shows reduced activity compared to wild-type DSR2 (Fig. 4D), indicating that NADase activity may be enhanced by DSR2 tetramerization. Our results highlight the importance of SIR2 assembly for NADase activity in the DSR2 system.

**Structural basis for viral inhibition of DSR2 anti-phage defense**
Under evolutionary pressure of the long-term race between bacteria and phages, phages encode evasion proteins that specifically inactivate the DSR2 phage defense system. Using a phage mating method, Gerb et al. revealed that the phage *Siphoviridae* SPbeta expresses anti-DSR2 protein DSAD1 to evade the DSR2 immune system of *Bacillus subtilis*[30]. The molecular mechanism of how DSAD1 inhibits the DSR2 system remains unclear. Here, we collected high-quality cryo-EM data to generate a three-dimensional map of the DSR2-DSAD1 complex with a nominal resolution of 3.49 Å (Fig. 5A, B, Supplementary Table 1, Supplementary Fig. 4), providing detailed information on the complex architecture and functional insights. The structure shows that DSAD1

binds directly to the CTD of DSR2 at the site typically used for Tube binding and activation of DSR2 (Fig. 5C). Structure and sequence alignments show that the inhibitory DSR2-DSAD1 complex cannot bind to the phage tube protein due to a steric clash between two proteins at the binding interface (Fig. 5D, Supplementary Fig. 6B). Our results demonstrate that DSAD1 competes with Tube for the binding site of DSR2, resulting in the inhibition of DSR2 activation by Tube. The DSAD1 protein and DSR2 assemble in a 2:4 ratio, forming an inhibited state. Each X-shaped DSR2 dimer interacts with one DSAD1 protein (Fig. 5E, Supplementary Fig. 10B). Notably, two DSAD1 proteins can be located either on the same side or on the diagonal side of the DSR2 tetramer (Supplementary Fig. 13). The interactions between DSR2 and DSAD1 are mainly contributed by hydrophobic interactions (F59 of DSAD1 to I918/W919/L922/Y962/L966 of DSR2(a)) (Fig. 5F) and further strengthened by hydrogen bonds (S107/S108 of DSAD1 to S957/N961 of DSR2(a)) (Fig. 5G). There is also an additional hydrophobic interaction between DSAD1 and the opposite DSR2 (L14/V15/Y16 of DSAD1 to F576/Y574 of DSR2(b)) (Fig. 5H). The DSR2 mutant Y574G/F576G does not interact with DSAD1 (Supplementary Fig. 14), and the DSAD1 mutant L14A/V15A/Y16A also loses the ability to interact with DSR2 (Supplementary Fig. 14). The in vitro NADase activity assay shows that the presence of DSAD1 significantly inhibited the activation of DSR2 by Tube (Fig. 5I). The residual activity observed could be due to dissociation of DSAD1 from DSR2, allowing DSR2 and Tube to form an active complex. Alternatively, Tube proteins could bind to empty sites in the DSR2-DSAD1 complex, resulting in partial activity.

**Tube-mediated conformational changes activate NADase activity of DSR2**
To elucidate the molecular mechanism by which Tube binding activates DSR2, we performed a comparative analysis of the structures

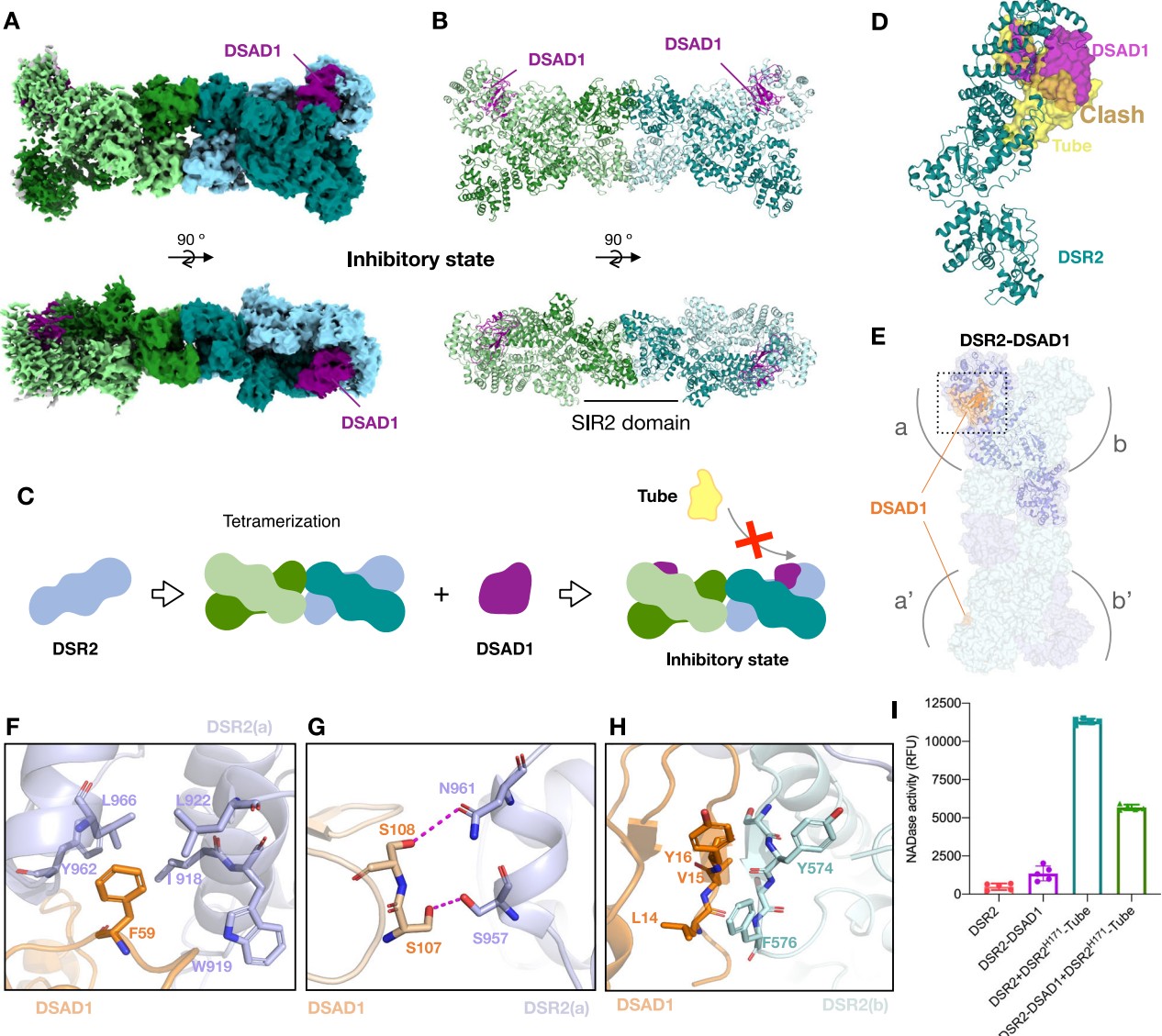

**Fig. 5 | Structural basis of viral inhibition of DSR2 defense.** Top and side views of the cryo-EM density map (**A**) and the atomic model (**B**) of the DSR2-DSAD1 complex. Four DSR2s are colored in pale green, forest, light blue, and teal; DSAD1 proteins are colored in purple. **C** Schematic model of DSR2-DSAD1 co-complex formation resulting in inhibition of NAD$^+$ hydrolysis by DSR2 due to absence of Tube protein. Same color code as in (**A**); Tube is shown in yellow. **D** Structural alignment of a DSAD1 (purple) and Tube (yellow) both binding to the same site on DSR2 (teal). The steric clash between DSAD1 and Tube protein occurs at the same binding interface of the DSR2 CTD. **E** Overall structure of the DSR2-DSAD1 complex, DSR2s are colored in blue and cyan; DSAD1 proteins are colored in orange. **F−H** Detailed insights into the interactions between DSR2 and DSAD1 protein. Two major regions of hydrophobic interactions and one major region of hydrogen bonding interactions between DSR2 and DSAD1. Key interacting residues are shown in stick representation. Hydrogen bonds are shown in black dashed lines. **I** In vitro NAD$^+$ degradation assays of wild-type DSR2 and the DSR2-DSAD1 complex. DSR2-DSAD1 significantly reduces NAD$^+$ cleavage activity. Data are presented as means ± SE ($n = 5$ independent experiments).

of DSR2 in the presence and absence of bound Tube, and in the DSAD1-bound inhibited state. In contrast to the "loose" structure of the apo DSR2 tetramer (~256 Å in diameter), the DSR2$^{H171A}$-Tube tetramer shows a more compact structure (~226 Å in diameter) (Fig. 6A). Furthermore, the DSR2-DSAD1 tetramer in the inhibited state shows a structure similar to the "loose" conformation observed in the inactive apo DSR2 tetramer (Fig. 6A). The presence of Tube contributes to this compact arrangement of the DSR2 tetramer. For each DSR2 protomer, binding of Tube proteins induces significant changes in the conformation and relative positions of the SIR2 and CTD domains, with an RMSD of 4.1 Å over 984 aligned Cα atoms (Fig. 6B). Upon interaction with the Tube protein, the circular solenoid lid of the CTD in DSR2 undergoes a downward rotation of approximately 30°, firmly attaching to the Tube. Moreover, the SIR2

domain undergoes a tilting motion of approximately 13° (Fig. 6B, Supplementary Movie 1). In contrast, binding of the inhibitor DSAD1 to DSR2 induced minor conformational changes in DSR2 (Fig. 6B). Therefore, it is reasonable to speculate that the conformational changes induced by Tube binding act as a trigger for the activation of DSR2 NADase activity.

To investigate how the conformational changes induced by Tube binding are transmitted to the SIR2 domain, we performed an analysis of key sites that could affect the SIR2 domain. In the tetrameric structure, each SIR2 domain is connected to the MD within the same protomer by a linker (aa 298−310). In addition, due to the formation of an X-shaped dimer where the two protomers cross, the SIR2 domain also forms significant interactions with the MD of the other protomer in the dimer (Fig. 6C). The loop (W143-Y148) of

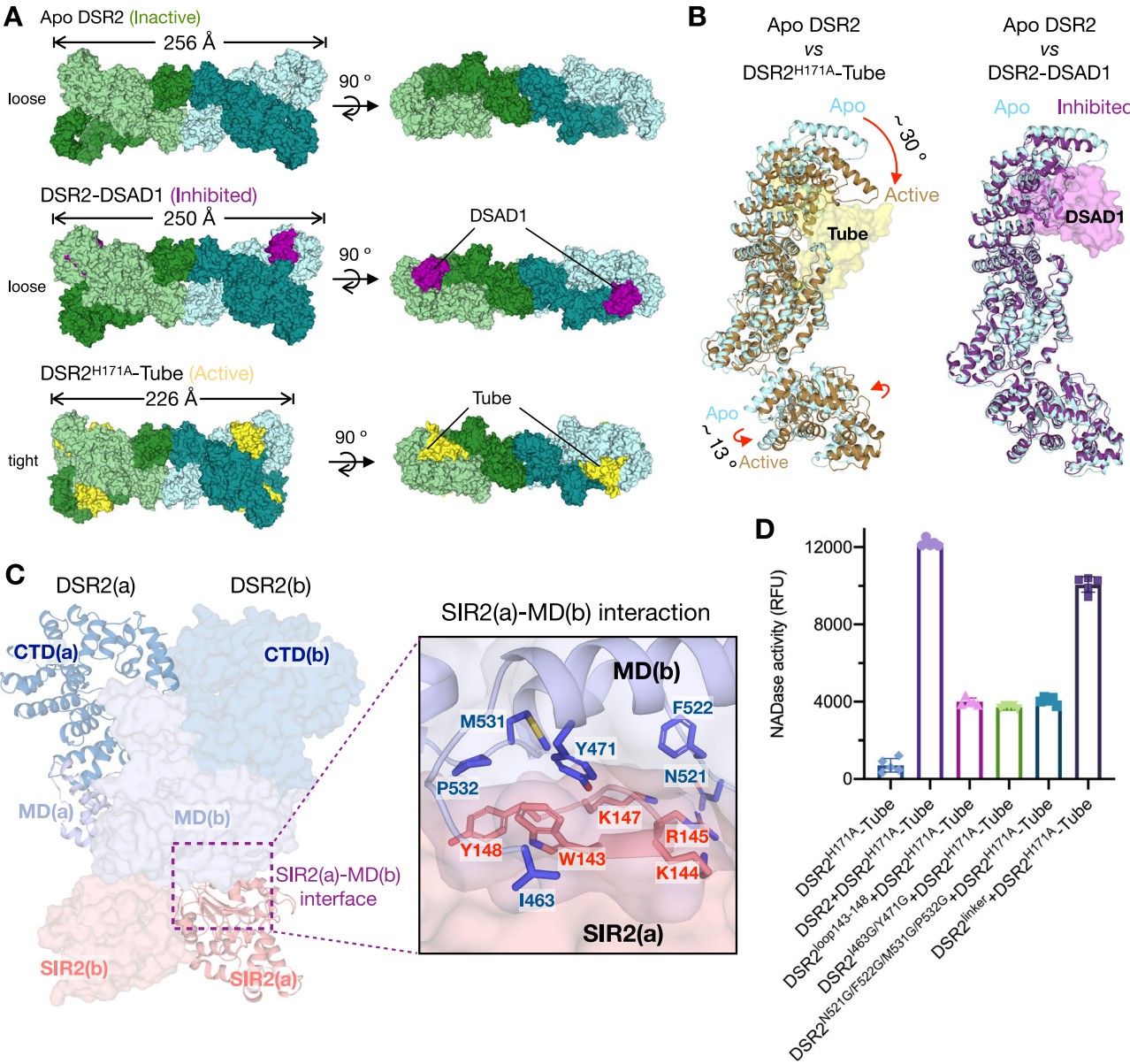

**Fig. 6 | Tube-mediated conformational changes activate NADase activity of DSR2. A** Top and side views of cryo-EM density maps of the apo DSR2 tetramer in the inactive state (top), the DSR2-DSAD1 complex in the inhibited state (middle), and the DSR2^H171A-Tube in the active state (bottom). Four DSR2s are colored in pale green, forest, light blue and teal, respectively. DSAD1 is colored in purple. Tube is colored in yellow. The conformation of the active state of DSR2^H171A-Tube was significantly more compact than the inactive and inhibited states. **B** Left, structural alignment of the protomers of apo DSR2 (light blue) and the active DSR2-Tube complex (brown for active DSR2 and yellow for Tube). Upon binding to the Tube protein, CTD circular solenoid lid of active DSR2 tilts ~30° to bind tightly to the Tube protein, while the SIR2 domain tilts ~13°. Right, structural alignment of the protomers of apo DSR2 (light blue) and DSR2-DSAD1 complex (purple for inhibited DSR2 and violet for DSAD1). **C** The interface between SIR2(a) and MD(b) from the adjacent DSR2(b). **D** The DSR2^loopI43-148, DSR2^I463G/Y471G, DSR2^N521G/F522G/M531G/P532G and DSR2^linker mutations at the SIR2(a)-MD(b) interface resulted in alterations in NAD⁺ cleavage. Wild-type DSR2 was used as control. Data are presented as means ± SD (*n* = 5 independent experiments).

SIR2(a) extends into the pocket of MD(b), which is composed of N521/F522/M531/P532/I463 and Y471 (Fig. 6C). We performed mutation analysis on the linker between SIR2(a) and MD(a) and the interface between SIR2(a) and MD(b). The results indicate that replacing the linker with a repeated GS sequence (DSR2^linker) has little effect on the activation of DSR2 by Tube (Fig. 6D). However, when the interface between SIR2(a) and MD(b) is disrupted, either by replacing loop 143–148 with a GS sequence (DSR2^loopI43-148) or by mutations in the MD pocket (DSR2^N521G/F522G/M531G/P532G and DSR2^I463G/Y471G), the activation of DSR2 by Tube is significantly affected (Fig. 6D). These results suggest that the interface between SIR2(a) and MD(b) is critical for the activation of DSR2 by Tube. The

signal generated by the conformational changes induced by Tube binding is likely transmitted through interactions between dimer molecules rather than within the same molecule, affecting the SIR2 domain and consequently activating its NADase activity.

## Discussion

Bacterial immune defense systems are types of pattern recognition receptors (PRRs) that can directly recognize the specific nucleic acids, proteins, or molecular structures of bacteriophage[3,4,10]. The recent surge of interest in bacterial pattern recognition receptors has led to remarkable discoveries of various bacterial immune systems that directly recognize phage-encoded proteins and initiate

**Fig. 7 | Schematic illustration of the NADase activation mechanisms of the DSR2 anti-phage system.** DSR2 tetramerizes to form a supramolecular complex that specifically recognizes phage tail tube proteins. This recognition leads to cellular NAD⁺ depletion and triggers an ineffective bacterial infection response. To overcome this problem, some phages encode the anti-DSR2 protein DSAD1, which blocks DSR2 activation by the tail tube, thereby preventing abortive infection and promoting phage replication.

cell death mechanisms as a defense strategy against phage invasion[1,2,6,7,9,30]. Upon recognition of a tail tube protein, the DSR2 system depletes NAD⁺, leading to a proposed abortive infection[30]. In this paper, we report the cryo-EM structures of the apo DSR2 tetramer and DSR2 in complex with tail tube protein and DSAD1, respectively. The structural and biochemical results provide several insights into the molecular mechanisms underlying the bacterial DSR2 immune system and the structural basis of viral immune evasion.

Based on the structural analysis, we propose the following model for a DSR2 system (Fig. 7). Without phage infection, the apo DSR2 tetramer in cells adopts an inactive, loose conformation with its N-terminal SIR2 domain in a self-suppressed state. Upon infection, the phage SPR tail tube protein is synthesized and binds to the middle domain (MD) and C-terminal domain (CTD) of DSR2. This interaction induces significant conformational changes in both the MD and CTD, resulting in a compact DSR2 conformation. The signal generated by the conformational change is likely transmitted through the MD to the SIR2 of the other protomer involved in the dimer. This process may activate the NADase activity of DSR2 by increasing the stability of the SIR2 or by facilitating the entry of substrate into the active site of DSR2. In this bound state, DSR2 can hydrolyze NAD⁺, resulting in a failed infection and preventing the phage from spreading throughout a population of cells. Remarkably, certain phages employ a strategy to evade the bacterial DSR2 system by encoding the inhibitory protein DSAD1. Upon infection, the phage SPbeta produces the DSAD1 protein, which interacts with the apo DSR2 tetramer, preventing binding of the tube protein and preventing activation of DSR2. This strategy effectively prevents abortive bacterial infection, allowing the phage to replicate. These phage-encoded proteins play critical roles in bacteriophage-host interactions by either activating or suppressing the bacterial immune system. Interestingly, DSR2 proteins from different species were not conserved in the activator binding regions, as well as in the DSR anti-defense binding regions (Supplementary Fig. 6B), indicating that recognition of phage proteins by DSR2 from different bacteria is species-specific, which is also consistent with species-specific features of bacteriophage infection. Our research extends our understanding of the mechanism by which the phage tail tube protein triggers NADase activation of the DSR2 system. DSAD1 hijacks the DSR2 system to aid phage envelopment, revealing the structural mechanism for evading host defenses.

## Methods

### Generation of constructs
The nucleotide sequence of the DSR2 from *Bacillus subtilis* (NCBI protein accession WP_128992496.1), the phage tail tube protein from *Siphoviridae* SPR (NCBI protein accession WP_010328117.1), and the anti-DSR2 protein DSAD1 from *Siphoviridae* SPbeta (NCBI protein accession WP_004399562.1) were synthesized and codon-optimized for expression in *E. coli*. The DSR2 gene and the mutants were cloned into the pCDFDuet vector (Novagen) linearized by BamHI and XhoI. The Tube and DSAD1 genes were cloned into the modified pRSF-Duet-1 vector (Novagen) with a cleavable N-terminal His6-SUMO tag linearized by BamHI and XhoI. All oligonucleotides used in this report are listed in Supplementary Table 2. All constructs were confirmed by DNA sequencing.

### Protein expression and purification
Vector expression was carried out in *E. coli* BL21(DE3) cells that were grown aerobically at 37 °C in lysogeny broth (LB) medium containing 50 μg/mL streptomycin or kanamycin. His6-SUMO-tagged (Tube and DSAD1) and His-tagged (DSR2) constructs were transfected into *E. coli* BL21 (DE3) cells (Novagen). The cells were cultured at 37 °C until OD600 reaches 0.6–0.8, and were induced by 0.5 mM isopropyl-β-D-1-thiogalactopyranoside (IPTG) at 25 °C for 16 h with constant shaking. Cells were collected and lysed by sonication in lysis buffer (25 mM Tris-HCl, pH 7.5, 20 mM imidazole, and 150 mM NaCl). Lysates were centrifuged at 30,000 × g for 45 min at 4 °C and the supernatant was purified using Ni-NTA beads (Qiagen). The column was washed with lysis buffer and the target proteins were eluted with lysis buffer containing 500 mM imidazole. The His6-SUMO tag from DSAD1 was removed by overnight Ulp1 protease digestion at 4 °C, followed by gel filtration on a G200 column (Superdex 200 Increase 10/300 GL) equipped with the storage buffer containing 25 mM Tris-HCl, pH 7.5, 150 mM NaCl. The His6-SUMO-Tube was directly purified using gel filtration on a G200 column equipped with the storage buffer. The final sample was concentrated and stored at −80 °C before use. To prepare the DSR2^H171A-Tube and DSR2-DSAD1 complexes, *E. coli* BL21(DE3) cells containing DSR2^H171A and HisSUMO-Tube (or HisSUMO-DSAD1) vectors were cultured at 37 °C until OD600 reaches 0.6–0.8, and were induced by 0.5 mM IPTG at 25 °C for 16 h with constant shaking. For DSR2^H171A-Tube complex, the protein sample was fractionated over the Hitrap Q column with a linear NaCl gradient from 50 mM to 1 M. All mutants and complexes

were purified by following a similar protocol to wild-type proteins. Peak fractions containing pure proteins were pooled and concentrated for cryo-EM.

## Fluorescence-based NADase assay

In the fluorescence-based NADase assay, ε-NAD⁺ (Nicotinamide 1, N6-ethenoadenine dinucleotide, BIOLOG Life Science Institute, BLG-N010) was used as the substrate. Hydrolysis of ε-NAD⁺ generates fluorescent ε-ADP-ribose, which was monitored by measuring the increase in fluorescence at 410 nm. These assays were performed in a 96-well microplate. Briefly, the 100 μl reaction system contains 25 mM Tris-HCl, pH 7.5, 150 mM NaCl, 500 μM ε-NAD⁺, 250 mM DSR2 or its mutants, and 250 mM Tube protein or its other varieties. The mixture was incubated at 25 °C for 40 min. Fluorescence emission at 410 nm was read using Multimode Plate Readers (PerkinElmer) after excitation at 300 nm. Statistical analyses were performed using GraphPad Prism software.

## Cryo-EM sample preparation and data collection

Purified apo DSR2 complex, DSR2-DSAD1 and DSR2$^{H171A}$-Tube complexes of 3.5 μL at a concentration of 1.5 mg/mL, 1.2 mg/mL, and 2 mg/mL, respectively, were applied to a glow-discharged Quantifoil grid (R 1.2/1.3 400 mesh, Au, Electron Microscopy Sciences), blotted for 5 s in 100% humidity at 4 °C and plunged into liquid ethane using a Vitrobot Mark IV (Thermo Fisher Scientific). For DSR2$^{H171A}$-Tube complex, 1 mM NAD⁺ was added before Cryo-EM sample preparation. Data collection of the cryo-EM datasets of DSR2 and DSR2-DSAD1 were performed on a 300 kV Titan Krios electron microscope (FEI) equipped with K3 and K2 Summit camera (Gatan) respectively, and a GIF Quantum energy filter operated with a slit width of 20 eV. All cryo-EM super-resolution micrographs were collected automatically using the Serial-EM package, yielding an image stack with a pixel size of 1.07 Å and 1.04 Å, respectively. The cryo-EM datasets of DSR2$^{H171A}$-Tube with NAD⁺ were collected with SerialEM42 on a Talos Arctica 200 kV FEG (Thermo Fisher Scientific) with a K2 summit direct electron elector (Gatan) and a GIF quantum energy filter (Gatan). The pixel size was calibrated at 1.0 Å under super-resolution mode. All the images were recorded at a defocus range of −1.2 mm to −1.8 mm, with a total electron dose of 60 electrons per Å² over 32 movie frames during a total exposure time of 4 s. Images were recorded by beam-image shift data collection methods. Each movie stack was motion-corrected by MotionCor2[40]. The dose-weighted micrographs were kept for further image processing using CryoS-PARC v4.3.0[41].

## Cryo-EM image processing

After data collection, the exact defocus value and contrast transfer function (CTF) of each micrograph were estimated using CryoSPARC's patch CTF estimation tool. Particles were automatically picked using Blob picking and Templet picking. For the data of apo DSR2 tetramer, a total of 457,033 particles were auto-picked from 649 micrographs. After 2D classification, 132,113 particles with good features were kept for further data processing (Supplementary Fig. 2). Three rounds of 3D classification were performed with a model created in CryoSPARC and low-pass filtered to 60 Å as a reference map, generating the best class containing 80,223 particles for further 3D refinement with C1 symmetry, yielding a final density map of 4.75 Å. To improve the density map, focus refinements were performed and reached a final resolution of 4.15 Å. Resolution is estimated based on the gold-standard Fourier shell correlation (FSC) with 0.143 criterion. For the data of DSR2$^{H171A}$-Tube complex, similar to apo DSR2 tetramer, a total of 943,799 particles were auto-picked from 2459 micrographs. After 2D classification, 298,712 particles with good features were kept for further data processing (Supplementary Fig. 3). Three rounds of 3D classification were performed with a model created in CryoSPARC and

low-pass filtered to 60 Å as a reference map, generating the best class containing 89,565 particles for further 3D refinement with C1 symmetry, yielding a final density map of 3.58 Å. Resolution is estimated based on the gold-standard Fourier shell correlation (FSC) with 0.143 criterion. To improve the density map, focus refinements were performed for the Tube and reached a final resolution of 3.08 Å. For the data of DSR2-DSAD1 complex, similar to apo DSR2 tetramer, a total of 629,718 particles were auto-picked from 2482 micrographs. After 2D classification, 292,132 particles with good features were kept for further data processing (Supplementary Fig. 4). Three rounds of 3D classification were performed with a model created in CryoSPARC and low-pass filtered to 60 Å as a reference map, generating the best class containing 105,782 particles for further 3D refinement with C1 symmetry, yielding a final density map of 3.86 Å. To improve the density map, focus refinements were performed and reached a final resolution of 3.49 Å. Resolution is estimated based on the gold-standard Fourier shell correlation (FSC) with 0.143 criterion.

## Model building and analysis

For the atomic model of apo DSR2 tetramer, DSR2$^{H171A}$-Tube, and DSR2-DSAD1 complexes, the structure of DSR2, Tube and DSAD1 predicted by AlphaFold2[42], as the initial model, were fitted into the cryo-EM maps using UCSF Chimera[43]. The resulting model was then manually rebuilt in COOT. PHENIX was used to refine the model against the cryo-EM density in real space and to ensure proper geometry. The model stereochemistry was evaluated using the comprehensive validation (cryo-EM) utility in PHENIX[44]. Structural figures were generated using PyMOL (The PyMOL Molecular Graphics System, Version 2.0 Schrödinger, LLC.) and ChimeraX[43]. Model versus map FSC curves were calculated using Phenix to obtain an estimate of the final resolution for the models (Supplementary Table 1). Pairwise structure comparison was performed on the Dali server.

## Toxicity assays

For toxicity assays of Tube-mediated DSR2 activation, His6-SUMO tagged Tube and untagged DSR2 (or H171A mutant) constructs were co-transformed into *E. coli* BL21 (DE3) cells (Novagen) and bacterial growth was monitored in LB agar plates with 0.05 mM IPTG.

## In-vivo pull-down assays

In-vivo pull-down assays to detect DSR2-Tube or DSR2-DSAD1 interactions were based on the co-expression system described above. For pulldown experiments of DSR2-Tube, 1 L of bacteria expressing both the His6-SUMO tagged Tube (or mutants) and DSR2$^{H171A}$ (or mutants) without the tag were co-expressed in *E. coli* BL21 (DE3) cells. Then the samples were immunoprecipitated using Ni-NTA beads and eluted in 25 mM Tris-HCl (pH 7.5), 150 mM NaCl and 500 mM imidazole. The eluted samples were determined by SDS-PAGE. The same method was used for the DSR2-DSAD1 pull-down experiments.

## Reporting summary

Further information on research design is available in the Nature Portfolio Reporting Summary linked to this article.

# Data availability

The cryo-EM density maps have been deposited into the Electron Microscopy Data Bank (EMDB) under accession numbers EMD-37610 (apo DSR2 tetramer), EMD-37606 (DSR2$^{H171A}$-Tube), and EMD-37607 (DSR2-DSAD1). The atomic coordinates have been deposited at the Protein Data Bank (PDB) under accession numbers 8WKX (apo DSR2 tetramer), 8WKS (DSR2$^{H171A}$-Tube), and 8WKT (DSR2-DSAD1). The source data underlying Figs. 2A, 4D, 5I and 6D and Supplementary Figs. 1, 5, 7, 11,12 and 14 are provided as a Source Data file. Other data are available from the corresponding author upon request. Source data are provided with this paper.

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

## Acknowledgements

Cryo-EM data collection was carried out at the Center for Biological Imaging, Core Facilities for Protein Science at the Institute of Biophysics, Chinese Academy of Sciences. We thank B. Zhu, X. Huang, G. Ji, X. Li, F. Sun, and other staff members at the Center for Biological Imaging for their support in data collection. We thank the Biological and Medical Engineering Core Facilities of the Beijing Institute of Technology for supporting experimental equipment. The project was funded by the National Natural Science Foundation of China (32171219 to A.G. and 32100757 to J.H.).

## Author contributions

J.H. and K.Z. generated most of the data presented here. Y.Gao, F.Y., Z.L., Y.Ge, and S.L. assisted with the experiments. J.H., J.Y., and A.G. designed the experiments and analyzed the data. J.H., and A.G. wrote the manuscript with contributions from all authors.

## Competing interests

The authors declare no competing interests.
