## [Peer Review File · Nature Communications]

Molecular basis of bacterial DSR2 anti-phage defense and viral immune evasionREVIEWER COMMENTS

Reviewer #1 (Remarks to the Author):

See attachment.

The manuscript entitled “Molecular basis of bacterial DSR2 anti-phage defense and viral immune evasion” by Huang et. al. presents cryo-EM structures of *Bacillus subtilis* DSR2 in three different states, elucidating the mechanisms governing DSR2 assembly, activation, and inhibition. DSR2, an NADase, can trigger bacterial cell death by depleting NAD⁺ when activated by a phage protein called SPR tail tube. Through structural and biochemical analyses, the authors showed that DSR2 activation is initiated by conformational changes triggered by its interaction with SPR tail tube. In addition, DSAD1, another phage effector protein, can inhibit the activity of DSR2 by direct binding, locking DSR2 in an auto-inhibited state. Notably, DSAD1 competes with SPR tube protein for the same binding site on DSR2. In summary, the manuscript provides robust evidence to support its major findings. However, some minor revisions are required before acceptance for publication.

1. In Fig. 5E, the addition of DSR2-DSAD1 into DSR2H171A-Tube promoted the NADase activity of DSR2, albeit not as effectively as wild-type DSR2. There are two potential explanations to this observation. First, DSAD1 may dissociate from DSR2, allowing DSR2 and Tube to form an active complex. Second, tube proteins may bind to the empty pockets in DSR2-DSAD1, thereby activating it. The authors are encouraged to clarify which hypothesis explains the activation of DSR2 when DSR2-DSAD1 and DSR2H171A-Tube were mixed together.
2. In Extended Data Fig.5, it would be beneficial to include a comparison of the gel filtration profiles of wild type DSR2-Tube and DSR2I259S/Y260G-Tube, rather than comparing wild type DSR2 and DSR2I259S/Y260G alone. In addition, please include an SDS-PAGE gel to confirm the binding capability of DSR2I259S/Y260G with Tube proteins.
3. The authors are advised to revise the manuscript to enhance its conciseness and precision. Below are a few examples.
 - a) In page 4 line 82-83, “The DSR2 subunits oligomerize into an autoinhibited homo-tetrameric assembly that appears as a 480 kDa bone-shaped supramolecular complex with four copies of the DSR2 protein” can be simplified as “Four DSR2 subunits assemble into an autoinhibited homo-tetrameric assembly, forming a ~480 kDa bone-shaped supramolecular complex”.
 - b) Page 5 line 106, “with a difference of 2.6 Å” should be “with a root-mean-square deviation of 2.6 Å”
 - c) Page 5, line 112, “inactive, silent conformation of the DSR2” should be “inactive conformation of the DSR2”
 - d) Page 5, line 115, “SIR2 domains in bacterial phage defense systems” should be “SIR2 domains in bacterial anti-phage defense systems”
 - e) Page 6, line 144-145, the sentence of “The four DSR----- to form a ~600 kDa supramolecular complex” is redundant to the prior sentence. I would suggest deleting this sentence.

- f) Page 6, line 148, “which immediately shows that only” should change to “indicating that only”
- g) Page 10, line 275, “conformation to stabilize its NADase active.” should be “conformation to stabilize its NADase active site.”

There are some other grammatic errors that I did not elaborate here.
Please go through the manuscript carefully and correct them.

REVIEWER COMMENTS

Reviewer #2 (Remarks to the Author):

Dear editor, dear authors,

In their manuscript, Huang and Zhu et al. have investigated the structural mechanism underlying catalytic activation of DSR2 systems and DSR2 inhibition by DSAD1. To this end, they have resolved various cryo-EM structures of DSR2 tetramers and DSR2 tetramers bound to Tube proteins and DSAD1 inhibitors. Based on these structures and various biochemical experiments, the authors propose a mechanism for catalytic activation of DSR2 (and inhibition thereof).

The authors provide a straightforward structural analysis with clear figures and textual description. I have some major comments regarding the validation of the structural models that I suggest to address, as well as various minor comments

I hope that the authors find my comments useful and that they can use them to enhance their exciting findings.

Major comments

-While the authors nicely show (and describe) how the structures allow identification of residues important for DSR2 dimerization, tetramerization, and interactions with the tube monomers and DSAD1, there is no experimental validation of most of these interactions. Especially given the poor resolution of some structures, which most likely does not allow proper building of the model at certain places, it cannot directly be deduced that these are indeed the interacting residues. Point mutants should be made and constructed to verify whether described residues important for DSR2 dimerization, and interactions with the tube monomer, and DSAD1 indeed are important and disturb the described interactions (as the authors already have done nicely for the NAD interacting residues and tetramerization surfaces). This is essential to validate their structural observations and to strengthen their conclusions.

-164: this is speculative. Could the authors make mutations in which such movements are inhibited and NADase activity is lost? The final answer to how Tube binding catalytically activates the SIR2 domains is currently unclear, which is a shame.

Minor comments:

-It would be insightful to show that co-expression of DSR2 and the Tube protein is toxic in E. coli (and that co-expression of the mutant is not).

-134: this is speculative - could it also be that DSR2H171A protomers dissociate and interact with wt DSR2 protomers and therefore that becomes activated? Can you show dissociation of NAD? The authors should verify what happens here.

-It would be insightful (and corroborating conclusions drawn by the authors) to do a sequence conservation analysis - how conserved are the residues indicated to be important for protein-protein and protein-NAD interactions as claimed by the authors?

Typos/clarification:

-Some of the text is a bit redundant. For example, in the introduction it is mentioned several times that recognition of phage proteins is important, but that mechanisms are unknown.

-28: this is not the mechanism, this is the PhAMP. The mechanism would include (in this case) triggering Abi.

-27: Which type of viral proteins? Edit: I now see this is described a couple sentences below. Flow could be improved here.

-61: systems

-84: it would be useful to refer to the DSR2 (tetrameric) complex and/or DSR2 protomers in this section to clarify what is being discussed. E.g. in sentence 87, is is not entirely clear what the authors mean. Do they think that DSR2 subunit alone (i.e. in absence of the tetramer) can mediate NADase activity?

-96: What is the helical solenoid domain? Ref?

-99: this suggests that also ThsA is activated by phage tails which is not true.

-106: What is this value? RMSD? Over how many residues? How is it determined?

-108: they are not the same, they are homologous.

-121: of which complex? DSR2?

-133: this suggests that the NAD⁺ has depletion activity. You can detect the depletion of NAD, or NADase activity.

-266: I would not say alone, as it is not clear that DSR2 forms a tetramer.

-Extended data figure 1: Falimant instead of filament

-Extended data figure 4: would be nice to include RMSD scores in the figure.

We thank the reviewers for their valuable comments. In the revised manuscript, we have performed more experiments and carefully addressed the reviewers' concerns. We believe that the revised version, which takes into account suggestions of the reviewers and recommendations of the editor, is well suited for *Nature Communications*. Our detailed responses to reviewers' critiques are presented below (reviewers' comments are in red and our responses are in black).

Referee #1 (Remarks to the Author):

In their manuscript, Huang and Zhu et al. have investigated the structural mechanism underlying catalytic activation of DSR2 systems and DSR2 inhibition by DSAD1. To this end, they have resolved various cryo-EM structures of DSR2 tetramers and DSR2 tetramers bound to Tube proteins and DSAD1 inhibitors. Based on these structures and various biochemical experiments, the authors propose a mechanism for catalytic activation of DSR2 (and inhibition thereof). The authors provide a straightforward structural analysis with clear figures and textual description. I have some major comments regarding the validation of the structural models that I suggest to address, as well as various minor comments.

We thank the reviewer for summarizing the key contributions of our work and finding this study "straightforward" and "clear figures and textual description". By following the reviewer's suggestions, we have conducted considerably more experiments to address the concerns.

Major comments

1. While the author nicely show (and describe) how the structures allow identification of residues important for DSR2 dimerization, tetramerization, and interactions with the tube monomers and DSAD1, there is no experimental validation of most of these interactions. Especially given the poor resolution of some structures, which most likely does not allow proper building of the model at certain places, it cannot directly be deduced that these are indeed the interacting residues. Point mutants should be made and constructed to verify whether described residues important for DSR2 dimerization, and interactions with the tube monomer, and DSAD1 indeed are important and disturb the described interactions (as the authors already have done nicely for the NAD interacting residues and tetramerization surfaces). This is essential to validate their structural observations and to strengthen their conclusions.

We thank the reviewer for the suggestions and agree with the reviewer that it is important to clearly identify the residues involved in the interactions. The cryo-EM structures of the DSR2^{H171A}-Tube and DSR2-DSAD1 were locally refined and determined to 3.08 Å (Figure. R1-A) and 2.93 Å (Figure. R1-B), respectively. The quality of the density maps allows us to

build accurate models for the interaction interface of both DSR2-Tube and DSR2-DSAD1. The density maps of key residues on the interaction interface are shown in Figure. R1.

A DSR2^{H171A}-TUBE (local refine)

B DSR2-DSAD1 (local refine)

Figure R1. Local resolutions of the DSR2^{H171A}-Tube (A) and DSR2-DSAD1 (B). Cryo-EM densities for the key interacting residues are shown. (Supplementary Figure 11 in the revised version)

While the density map provides sufficient confidence to identify key participating residues, we fully agree with the reviewer's suggestion that it is essential and important to verify the residues involved in DSR2 dimerization, tetramerization, and interactions with the Tube monomers and DSAD1 by constructing point mutants. Following the reviewer's suggestion, we have performed additional experiments.

(1) Validation of key interactions of DSR2 dimerization and tetramerization: To identify the key residues governing DSR2 dimerization and tetramerization, we introduced glycine or serine substitutions for crucial amino acids involved in these interactions. The aggregation status of

mutants was then assessed using size exclusion chromatography (SEC). The mutant DSR2^{W143G/Y148G/Y552G/V556S/F559G/N563S}, with disruption at the dimeric interface, behaves as monomer, whereas the DSR2^{I259S/Y260G} mutant, with disruption at the tetrameric interface, behaves as a dimer (Figure R2).

Figure R2. Size-exclusion chromatography of wild-type DSR2, DSR2^{I259S/Y260G} and DSR2^{W143G/Y148G/Y552G/V556S/F559G/N563S}.

(2) Validation of DSR2 interactions with Tube monomer and DSAD1: To confirm the key amino acids involved in the interactions of DSR2 with Tube monomer and DSAD1, we first tried an *in vitro* His-tag pull-down assay using the individually purified DSR2 (N-terminal His-tag), Tube (no tag), and DSAD1 (to tag). However, the results indicate that the purified Tube protein cannot be pulled down (Figure R3-A). Cryo-EM analysis reveals that the purified Tube protein exists in the form of stable cyclic hexamers (Figure R3-B). Based on the observation of monomeric Tube protein in the cryo-EM structure of DSR2^{H171A}-Tube, we hypothesized that the purified hexameric Tube protein hinders the interaction with DSR2. In addition, the individually expressed DSAD1 protein exists predominantly in insoluble inclusion bodies. Therefore, the properties of individually expressed and purified Tube and DSAD1 proteins are not suitable for *in vitro* verification of DSR2-Tube and DSR2-DSAD1 interactions.

Figure R3. *In vitro* pull-down assays of the DSR2 and Tube proteins. A. DSR2 does not bind purified Tube protein. Representative of two replicates. B. The cryo-EM image of the purified Tube protein shows that Tube is in the form of stable cyclic hexamers.

On the other hand, we can successfully obtain the DSR2^{H171A} (no tag)-Tube (His-SUMO tag) complex (Figure. R4) and the DSR2 (no tag)-DSAD1 (His-SUMO tag) complex (Figure. R5) using the *E. coli* co-expression system. Therefore, we employed the co-expression system and *in vivo* pull-down to investigate the key amino acids involved in the interactions of DSR2 with Tube monomer and DSAD1. As expected, either the DSR2 mutant H171A/L495S/L497G/L498S (Figure. R4-B) or the Tube mutant L50A/Y51A/I52A (Figure. R4-C), both with disruptions at the interaction surface, fail to form the DSR2-Tube complex. In the case of the DSR2-DSAD1 complex, the DSR2 mutant Y574G/F576G does not interact with DSAD1 (Figure. R5-B), and the DSAD1 mutant L14A/V15A/Y16A also loses the ability to interact with DSR2 (Figure. R5-C). Taken together, the co-expression results indicate the importance of these residues in the interaction of DSR2 with the Tube monomer and DSAD1.

Figure R4. DSR2^{H171A} without tag and Tube protein with His-SUMO tag were co-expressed in *E. coli* and purified with Ni-column to obtain the DSR2^{H171A} -Tube complex (A). The DSR2 mutant

H171A/L495S/L497G/L498S blocks its interaction with Tube (B). The Tube mutant L50A/Y51A/I52A cannot pull down DSR2^{H171A} (C).

Figure R5. DSR2 without tag and DSAD1 protein with His-SUMO tag were co-expressed in *E. coli* and purified with Ni-column to obtain the DSR2-DSAD1 complex (A). The DSR2 mutant Y574G/F576G did not interact with DSAD1 (B). DSAD1 with point mutations (L14A/V15A/Y16A) also lost the ability to interact with DSR2 (C).

2 164: this is speculative. Could the authors make mutations in which such movements are inhibited and NADase activity is lost? The final answer to how Tube binding catalytically activates the SIR2 domains is currently unclear, which is a shame.

We thank the reviewer for the suggestions. Indeed, the mechanism by which Tube binding induces the NADase activity of SIR2 domains is very interesting. To further investigate the mechanism, we followed the reviewer's suggestion and attempted to inhibit the movement of SIR by making mutations. Considering that the flexible linker (298-ENKFITKDDEVID-310) connecting the SIR2 domain and the MD domain may play an important role in the movement of the SIR2 domain, we try to restrict the movement of the SIR2 domain by replacing the flexible linker (aa 298-310) with various rigid linkers (predicted to be α -helices), including AAA, AAAAA, AAAAAA, ARSTLARSTLARS, and EAAAKEAAKAP, and then evaluate the effect of these changes on NADase activity. Regrettably, the mutants with replaced linkers are either not expressed or are expressed as insoluble inclusion bodies, further indicating the indispensable role of the flexible linker (aa 298-310) in preserving the structure and function of DSR2.

Minor comments:

3. It would be insightful to show that co-expression of DSR2 and the Tube protein is toxic in *E. coli* (and that co-expression of the mutant is not).

We thank the reviewer for the suggestion. We co-expressed wild-type DSR2 with Tube protein and monitored bacterial growth. As shown in Figure R6, co-expression of wild-type DSR2

and the Tube protein resulted in cellular toxicity, and this toxicity was alleviated when these DSR2 proteins harbored the H171A point mutation (Figure R6).

Figure R6. Co-expression of wild-type DSR2 with Tube protein induces cellular toxicity (left). Mutant DSR2^{H171A} exhibits reduced toxicity (right). (Supplementary Figure 8 in the revised version)

4. 134: this is speculative - could it also be that DSR2^{H171A} protomers dissociate and interact with wt DSR2 protomers and therefore that becomes activated? Can you show dissociation of NAD? The authors should verify what happens here.

We thank the reviewer for the question. We performed additional NADase assays and the results (Figure R7) showed that (1) apo wt-DSR2, apo DSR2^{H171A} and DSR2^{H171A}-Tube did not have NADase activity; (2) wt-DSR2 mixed with DSR2^{H171A} did not have NADase activity; (3) the mixture of wt-DSR2 with DSR2^{H171A}-Tube exhibited NADase activity. These results demonstrate that the presence of Tube proteins is required for DSR2 to exert NADase activity. In the mixture of wt-DSR2 with DSR2^{H171A}-Tube, the only possible source of Tube is dissociation from DSR2^{H171A}-Tube. In addition, given the small molecular weight difference between wild-type DSR2 and the H171A mutant, it is a real challenge to detect the dynamic changes in the complex and to determine the presence of hybrid tetramers consisting of both wild-type and mutant forms in solution, as well as to perform further studies to identify which tetramer exhibits enzyme activity. Taken together, we cannot completely exclude the possibility that the wild-type DSR2 is activated by the dissociated DSR2^{H171A}-Tube protomer. Therefore, we have revised the sentence to the following in the updated manuscript.

Lines 138-140, “We suspect that wild-type DSR2 could be activated either by the DSR2^{H171A}-Tube protomer dissociated from the tetrameric complex or by interaction with monomeric Tube released from the complex.”

Figure R7. *In vitro* NAD⁺ degradation assays of DSR2.

5. It would be insightful (and corroborating conclusions drawn by the authors) to do a sequence conservation analysis - how conserved are the residues indicated to be important for protein-protein and protein-NAD interactions as claimed by the authors?

We thank the reviewer for the suggestion. Following reviewer's suggestion, we performed sequence alignment of DSR2 from various bacterial species. Residues Asn133 and His171 within the NAD⁺-binding sites of DSR2 are conserved among other sirtuin proteins, such as ThsA, pAgo, DSR1, and HerA (Figure R8-A). The alignment of the full-length amino acid sequences of DSR2 revealed the conservation of residues Asn133 and His171 among different DSR2 systems in different bacterial species. In contrast, the binding sites for Tube and DSAD1 were not conserved (Figure R8-B). This observation suggests that recognition of phage proteins by DSR2 in different bacterial species is species-specific, consistent with species-specific features of bacteriophage infection. These sequence alignment results have been included in the revised manuscript.

Lines 109-111, "Sequence alignment showed that residues Asn133 and His171 within the NAD⁺-binding sites of DSR2 are conserved among ThsA and other sirtuin proteins, such as, pAgo, DSR1, and HerA (Supplementary Fig.5A).

Lines 288-292, "Interestingly, DSR2 proteins from different species were not conserved in the activator binding regions, as well as in the DSR anti-defense binding regions (Supplementary Fig.5B), indicating that the recognition of phage proteins by DSR2 from different bacteria is species-specific, which is also consistent with species-specific features of bacteriophage infection."

Figure R8. Multiple sequence alignment of sirtuin proteins and DSR2 from different species. A. Sequence alignment of NAD⁺-binding region of DSR2, ThsA, pAgo, DSR1 and HerA. Residues Asn133 and His171 of DSR2 are shown in the red frame. B. Sequence alignment of DSR2 from different species. The blue diamond represents the NAD⁺ binding sites, the salmon-colored square

represents the Tube binding sites, and the teal triangle represents the DSAD1 binding sites.
(Supplementary Figure 5 in the revised version)

Typos/clarification:

6. Some of the text is a bit redundant. For example, in the introduction it is mentioned several times that recognition of phage proteins is important, but that mechanisms are unknown.

Thank you. We have revised the redundant parts in the introduction.

Lines 32-33, “However, unlike nucleic acid recognition immune systems, the identities of phage proteins triggers and the sensing mechanisms remain largely unknown.” has been changed to “However, the mechanism by which phage-encoded proteins trigger the bacterial immune system remains largely unknown.”

Delete “However, the phage-encoded proteins involved in bacterial immunity are still largely unknown.” in the Introduction section.

7. 28: this is not the mechanism, this is the PhAMP. The mechanism would include (in this case) triggering Abi.

Thank you. We have changed “Another immune defense mechanism is sensing phage proteins” to “Another PhAMP that is recognized by bacteria and triggers antiphage defense is phage protein.” (Lines 28-29)

8. 27: Which type of viral proteins? Edit: I now see this is described a couple sentences below. Flow could be improved here.

Thank you. We have changed “The sensor domains of Avs recognize conserved folds of viral proteins, which activates the DNA endonucleases to abortive infection” to “The sensor domains of Avs recognize viral proteins with conserved folds that act as structural patterns and activate DNA endonucleases, thereby inducing abortive infection.” (Lines 37-39)

9. 61: systems

Thank you. “system” has been changed to “systems” (Line 61).

10. 84: it would be useful to refer to the DSR2 (tetrameric) complex and/or DSR2 protomers in this section to clarify what is being discussed. E.g. in sentence 87, is is not entirely clear what the authors mean. Do they think that DSR2 subunit alone (i.e. in absence of the tetramer) can mediate NADase activity?

Thank you. We have clarified this section by changing “As expected, NADase activity was not observed with DSR2 proteins alone in the fluorescent ϵ -NAD assay” to “As expected, in the absence of activator binding, the tetrameric DSR2 showed no NADase activity in the fluorescent ϵ -NAD assay” (Lines 83-84); “DSR2 subunit” has been changed to “Each DSR2 protomer” (Line 87).

11. 96: What is the helical solenoid domain? Ref?

Thank you. We have changed the description to make it easier to understand and added the reference. “The CTD is topologically analogous to the helical solenoid domain and forms a circular solenoid structure at the four corners of apo DSR2, which may have a role in substrate recognition.” has been changed to “The CTD adopts a horseshoe-shaped α -helical solenoid structure at the four corners of apo DSR2, which may function in substrate recognition³⁶.” (Lines 95-97)

Reference:

36. Fournier, D. et al. Functional and genomic analyses of alpha-solenoid proteins. PLoS One 8, e79894 (2013).

12. 99: this suggests that also ThsA is activated by phage tails which is not true.

Thank you. We have changed “Similar to ThsA, a well-studied NADase effector containing the SIR2 domain that is activated by the immune second messenger molecules cADPR produced by ThsB^{1,32}, DSR2 exerts NADase activity and mediates bacterial programmed death after binding with phage tail tube²⁸.” to “DSR2 exerts NADase activity and mediates bacterial programmed death after binding with phage tail tube²⁸. Another SIR2 domain-containing protein, ThsA, is an extensively studied NADase effector activated by the immune second messenger molecules cADPR produced by ThsB^{1,32}.” (Lines 99-102)

13. 106: What is this value? RMSD? Over how many residues? How is it determined?

Thank you. The value is RMSD over 274 aligned C α atoms. The structure comparison was performed on the Dali server. We have added more details in the revised manuscript. “The structures of the SIR2 domain exhibit high similarity between DSR2 and SeThsA, with a RMSD (Root-Mean-Square Deviation) of 2.6 Å over 274 aligned C α atoms when superimposed” (Lines 105-107) .

The method was added in “Model building and analysis”. Lines 395-396, “Pairwise structure comparison was performed on the Dali server.”

14. 108: they are not the same, they are homologous.

Thank you. “same” has been changed to “homologous” (Line108).

15. 121: of which complex? DSR2?

Thank you. This sentence has been changed into “To define the molecular mechanism of the activation of DSR2 by phage Tube protein, we first studied the structure of the DSR2-Tube complex” (Lines 123-124).

16. 133: this suggests that the NAD+ has depletion activity. You can detect the depletion of NAD, or NADase activity.

Thank you. “we can clearly detect the depletion activity of NAD⁺” have been changed into “we can clearly detect the depletion of NAD⁺” (Lines 137-138).

17. 266: I would not say alone, as it is not clear that DSR2 forms a tetramer.

Thank you. We have changed this sentence to “we report the cryo-EM structures of apo DSR2 and DSR2 in complex with tail tube protein and DSAD1, respectively” in lines 272-273.

18. Extended data figure 1: Falimant instead of filament

Thank you for pointing out the typos. They have been corrected.

19. Extended data figure 4: would be nice to include RMSD scores in the figure.

Thank you. The RMSD scores have been included.

Figure R9. Comparison of SIR2 domains between ThsA and DSR2. (Supplementary Figure 10 in the revised version)

Referee #2 (Remarks to the Author):

The manuscript entitled "Molecular basis of bacterial DSR2 anti-phage defense and viral immune evasion" by Huang et. al. presents cryo-EM structures of *Bacillus subtilis* DSR2 in three different states, elucidating the mechanisms governing DSR2 assembly, activation, and inhibition. DSR2, an NADase, can trigger bacterial cell death by depleting NAD⁺ when activated by a phage protein called SPR tail tube. Through structural and biochemical analyses, the authors showed that DSR2 activation is initiated by conformational changes triggered by its interaction with SPR tail tube. In addition, DSAD1, another phage effector protein, can inhibit the activity of DSR2 by direct binding, locking DSR2 in an auto-inhibited state. Notably, DSAD1 competes with SPR tube protein for the same binding site on DSR2. In summary, the manuscript provides robust evidence to support its major findings.

We thank the reviewer for summarizing the key contributions of our work. We also thank the reviewer for finding our work "provides robust evidence to support its major findings". In response to the reviewer's comments, additional experiments have been conducted, and the manuscript has been revised in accordance with the provided suggestions.

1. In Fig. 5E, the addition of DSR2-DSAD1 into DSR2H171A-Tube promoted the NADase activity of DSR2, albeit not as effectively as wild-type DSR2. There are two potential explanations to this observation. First, DSAD1 may dissociate from DSR2, allowing DSR2 and Tube to form an active complex. Second, tube proteins may bind to the empty pockets in DSR2-DSAD1, thereby activating it. The authors are encouraged to clarify which hypothesis explains the activation of DSR2 when DSR2-DSAD1 and DSR2H171A-Tube were mixed together.

We thank the reviewer for the discussion. In review's second hypothesis, it is expected that a ternary complex, comprising DSR2, Tube, and DSAD1, forms in solution. We modeled the DSR2-DSAD1 complex bound to the Tube without encountering steric hindrance, indicating that the formation of ternary complexes is structurally plausible (Figure R10-A). To experimentally verify whether DSR2, Tube, and DSAD1 can form a ternary complex, we performed the pull-down assay (Figure R10-B). DSR2^{H171A} (no tag) -Tube (His-SUMO tag) complex was mixed with DSR2(no tag) -DSAD1 (no tag) and then purified by Ni-column. The result shows that only DSR2 and Tube are eluted, but not DSAD1, suggesting that DSR2, Tube and DSAD1 cannot form a ternary complex. Considering that the pull-down assay may not fully simulate physiological conditions, we cannot completely exclude the possibility that the Tube monomer binds to the empty pockets in the DSR2-DSAD1 complex. Therefore, we have added the following discussion in the revised manuscript.

Lines 246-250, "Compared to DSR2H171A-Tube, the mixture of DSR2-DSAD1 and DSR2H171A-Tube shows partial NADase activity (Fig.5E). We speculate that DSAD1 might dissociate from DSR2, allowing DSR2 and Tube to form an active complex. Alternatively, Tube proteins could bind to the empty pockets in DSR2-DSAD1, activating the complex."

Figure R10. A. Structural model of DSR2-Tube-DSAD1 complex. B. DSR2, Tube, and DSAD1 cannot form a ternary complex in pull-down assay. Representative of two replicates.

2. In Extended Data Fig.5, it would be beneficial to include a comparison of the gel filtration profiles of wild type DSR2-Tube and DSR2^{I259S/Y260G}-Tube, rather than comparing wild type DSR2 and DSR2^{I259S/Y260G} alone. In addition, please include an SDS-PAGE gel to confirm the binding capability of DSR2^{I259S/Y260G} with Tube proteins.

We thank the reviewer for the suggestion. We have included the comparison of the gel filtration profiles of DSR2^{H171A}-Tube and DSR2^{H171A/I259S/Y260G}-Tube and also the SDS-PAGE analysis of DSR2^{H171A/I259S/Y260G}-Tube^{HisSUMOtag} in Supplementary Fig. 12 of the revised manuscript.

Figure R11. Size-exclusion chromatography analysis of wild-type DSR2, DSR2^{I259S/Y260G}, DSR2-Tube^{HisSUMOtag} and DSR2^{H171A/I259S/Y260G}-Tube^{HisSUMOtag}. SDS-PAGE analysis of DSR2^{H171A/I259S/Y260G}-Tube^{HisSUMOtag} was shown. (Supplementary Figure 12 in the revised version)

3. The authors are advised to revise the manuscript to enhance its conciseness and precision. Below are a few examples.

a) In page 4 line 82-83, “The DSR2 subunits oligomerize into an autoinhibited homotetrameric assembly that appears as a 480 kDa bone- shaped supramolecular complex with four copies of the DSR2 protein” can be simplified as “Four DSR2 subunits assemble into an

autoinhibited homo-tetrameric assembly, forming a ~480 kDa bone-shaped supramolecular complex”.

Thank you. We have revised the manuscript according to your suggestion.

b) Page 5 line 106, “with a difference of 2.6 Å” should be “with a root-mean- square deviation of 2.6 Å”

Thank you. “with a difference of 2.6 Å” has been changed into “with a RMSD (Root-Mean-Square Deviation) of 2.6 Å” (Line 106).

c) Page 5, line 112, “inactive, silent conformation of the DSR2” should be “inactive conformation of the DSR2”

Thank you. We have revised the manuscript according to your suggestion.

d) Page 5, line 115, “SIR2 domains in bacterial phage defense systems” should be “SIR2 domains in bacterial anti-phage defense systems”

Thank you. We have revised the manuscript according to your suggestion.

e) Page 6, line 144-145, the sentence of “The four DSR----- to form a ~600 kDa supramolecular complex” is redundant to the prior sentence. I would suggest deleting this sentence.

Thank you. We have deleted this sentence.

f) Page 6, line 148, “which immediately shows that only” should change to “indicating that only”

Thank you. We have revised the manuscript according to your suggestion.

g) Page 10, line 275, “conformation to stabilize its NADase active.” should be “conformation to stabilize its NADase active site.”

Thank you. We have revised the manuscript according to your suggestion.

There are some other grammatic errors that I did not elaborate here. Please go through the manuscript carefully and correct them.

Thank you for the suggestion. We had revised the manuscript and the grammatical errors had been revised.

We thank the reviewers again for their valuable comments and discussions, which really improve our manuscript.

In addition to the revisions based on the reviewers' suggestions, we also found and corrected a mistake in Figure 4D and the accompanying description in the manuscript.

In Figure. 4D of the original manuscript, we mistakenly used the DSR2^{H171A/I259S/Y260G}-Tube complex in these assays (in fact, the DSR2^{I259S/Y260G}-Tube complex should be used). Therefore, the change in NADase activity observed for the DSR2^{H171A/I259S/Y260G}-Tube cannot be attributed to the disruption of tetramerization, as the mutation in the catalytic site (H171A) confounds the interpretation. In the revised manuscript, we compared the NADase activity of wild-type DSR2 with that of the DSR2^{I259S/Y260G} mutant, which specifically disrupts the tetrameric interface. The result shows that DSR2^{I259S/Y260G} still reduces (but does not completely abolish) the activity, suggesting that tetramerization plays a role in enhancing the NADase activity of DSR2.

Based on these findings, we have made changes to Figure. 4D and the manuscript as described below. It is important to note that these changes do not affect the main conclusions drawn previously.

Lines 226-228,

“As expected, the I259S/Y260G mutant appeared as a dimer form according to the SEC analysis (Extended Data Fig.5) and the NADase activity was significantly decreased (Fig.4D), indicating that DSR2 NADase function requires DSR2 tetramerization.”

has been changed to

“As expected, the DSR2^{I259S/Y260G} appeared as a dimer form according to the SEC analysis (Supplementary Fig.12) and shows reduced activity compared to wild-type DSR2 (Fig.4D), indicating that the NADase activity may be enhanced by DSR2 tetramerization.”

Figure 4. The SIR2 assembly of the DSR2. **A.** Overview structure of SIR2 assembly in DSR2^{H171A}-Tube complex. **B.** The tetramerization of SIR2 domains. The dashed squares show the head-to-head contacts formed by SIR2 domains. **C.** Detailed insights into the head-to-head interactions. Key interacting residues are shown in stick representation. The I259 and Y260 in SIR2 $\alpha 4$ helix interact with the I90 and V94 in SIR2 $\alpha 12$. **D.** Mutation I259S/Y260G in the SIR2 assembly interface decreased NAD⁺ cleavage. Wild-type DSR2 was used as a control. NADase activity was decreased in the I259S/Y260G mutant. Data are presented as means \pm SD (n=5).

REVIEWER COMMENTS

Reviewer #1 (Remarks to the Author):

The authors have fully addressed my concerns. The manuscript can be published as it is.

Reviewer #2 (Remarks to the Author):

Dear editor, dear authors,

While the authors have addressed some of the issues that I have raised in their rebuttal, several of my points remain unaddressed in the manuscript. While the structures provide insightful leads, further mutational analysis should be described to support all claims. It is beyond my understanding why the authors do give (at least some of) this information in the rebuttal but not in the manuscript. Do they not trust the data themselves? Or do they think they do not support their claims?

Furthermore, the mechanism by which DSR2 gets activated in presence of the Tube (or in presence of the DSRH171A-Tube complex, and even in presence of the inhibitor) remain vague and speculative (or at least the description is unclear to me).

The manuscript can provide an interesting and important advance, but in its current state the manuscript feels unfinished.

Furthermore, there are numerous textual errors which make it more difficult to understand this manuscript. It would benefit from critical proofreading.

-29: Highlighted sentence: what phage protein?

-32: trigger what bacterial immune system? Most immune systems are not triggered by the phage-encoded proteins.

-84: E-NAD is not fluorescent, E-ADPR is. Better to rephrase this sentence for clarity.

-85: it might be more clear to refer to this complex as the apo DSR2 complex or apo DSR2 tetramer throughout this manuscript, to avoid confusion with monomeric DSR2/DSR2 protomers.

-88: clearly the domains are identifiable, they are described in the next sentence. Do the authors mean they have no known homology to other known domains?

-97: beyond just giving a reference here, the authors might provide the information that leads to this hypothesis (e.g. 'as observed for..')

-108: are conserved in both proteins

-119: this is not analogous, as the SIR2 domains are homologs, their activity is also homologous.

-135: NADase free is not very accurate: it is a catalytic mutant lacking NADase activity.

-145: I assume this is the mutant, not the WT?

-151: huge is a arbitrary term, remove

-197: this is not a hydrogen bond but a salt bridge?  the importance of this bond needs to be validated experimentally to support this claim.

-199-207: these claims need mutational analysis for support

-247: I do not understand how this inhibitor can activate the complex, this should be further clarified and/or verified.

We thank the reviewers for their valuable suggestions. In the revised manuscript, we have performed additional experiments to investigate the activation mechanism of DSR2 in the presence of Tube. In addition, we have expanded our mutational analysis to validate the interactions between molecules, covering almost all previously reported interactions. In response to the reviewer's feedback, we have systematically considered and implemented revisions to improve the clarity of the article. Changes in the manuscript are highlighted in yellow. We hope the revisions will address the reviewer's concerns and improve the overall quality of the manuscript. Our detailed responses to reviewers' critiques are presented below (reviewers' comments are in red and our responses are in black).

Reviewer #1

The authors have fully addressed my concerns. The manuscript can be published as it is.

We thank the reviewer for the positive comments.

Reviewer #2

While the authors have addressed some of the issues that I have raised in their rebuttal, several of my points remain unaddressed in the manuscript. While the structures provide insightful leads, further mutational analysis should be described to support all claims. It is beyond my understanding why the authors do give (at least some of) this information in the rebuttal but not in the manuscript. Do they not trust the data themselves? Or do they think they do not support their claims?

We appreciate the reviewer's valuable suggestions. These mutation results are robustly reproducible. We are glad to include these data in the revised manuscript.

Lines 77-82, “To identify the key residues governing DSR2 dimerization and tetramerization, we introduced glycine or serine substitutions for key amino acids involved in these interactions. The oligomerization status of the mutants was then assessed by size exclusion chromatography (SEC). The DSR2^{W143G/Y148G/Y552G/V556S/F559G/N563S} mutant with disruption at the dimeric interface behaves as a monomer, whereas the DSR2^{I259S/Y260G} mutant with disruption at the tetrameric interface behaves as a dimer (Supplementary Fig.5).”

Lines 150-152, “The DSR2^{L495S/L497G/L498S} and Tube^{F204A/M206A} mutants disrupt the interactions between DSR2 and Tube and fail to form the DSR2-Tube complex (Supplementary Fig.11).”

Lines 160-162, “Mutations at Site2 (Tube^{L50A/Y51A/I52A}) and Site3 (Tube^{S31A/F32A}), which disrupt the interaction surfaces, prevent the formation of the DSR2-Tube complex (Supplementary Fig.11).”

Lines 210-212, “The DSR2 mutant Y574G/F576G does not interact with DSAD1 (Supplementary Fig.14), and the DSAD1 mutant L14A/V15A/Y16A also loses the ability to interact with DSR2 (Supplementary Fig.14).”

Furthermore, the mechanism by which DSR2 gets activated in presence of the Tube (or in presence of the DSRH171A-Tube complex, and even in presence of the inhibitor) remain vague and speculative (or at least the description is unclear to me).

We thank the reviewer for raising this concern. We have observed that Tube binding induces significant conformational changes in DSR2, resulting in a more compact tetramer. We also performed further mutation experiments, which suggested that the signal generated by the conformational change is transmitted through the Tube-bound MD of one protomer to the SIR2 domain of the other protomer involved in the dimer. This process may activate the NADase activity of DSR2 by increasing the stability of SIR2 or by facilitating the entry of substrate into the active site of DSR2. However, it should be noted that, our cryo-EM structures did not capture every snapshot for conformational changes of the SIR2 domain during the activation process. In the revised manuscript, we have included this information in a new section (Tube-mediated conformational changes activate NADase activity of DSR2) and a new figure (Fig.6). In addition, we have discussed the mechanistic implications of this aspect in the Discussion section.

The manuscript can provide an interesting and important advance, but in its current state the manuscript feels unfinished.

Furthermore, there are numerous textual errors which make it more difficult to understand this manuscript. It would benefit from critical proofreading.

Thanks for the suggestion. We have systematically considered and implemented revisions (highlighted in yellow) to improve the clarity of the article.

-29: Highlighted sentence: what phage protein?

-32: trigger what bacterial immune system? Most immune systems are not triggered by the phage-encoded proteins.

We thank the reviewer for pointing out these issues. As detailed in the second paragraph of the original Introduction, we had extensively outlined some specific phage proteins responsible for triggering different bacterial immune systems. Based on the reviewer's valuable suggestions, we have further included additional immune systems activated by different phage proteins in

the revised manuscript. In addition, we have reorganized and combined the first and second paragraphs of the Introduction and simplified the description to improve overall clarity.

-84: ϵ -NAD is not fluorescent, ϵ -ADPR is. Better to rephrase this sentence for clarity.

Thanks for the suggestion. We have revised the manuscript and included additional details about this assay in the Methods section.

Lines 82-83, “As expected, in the absence of activator binding, DSR2 showed no detectable *in vitro* NADase activity.”

Lines 331-334, “In the fluorescence-based NADase assay, ϵ -NAD⁺ (Nicotinamide 1, N⁶-ethenoadenine dinucleotide, BIOLOG Life Science Institute, BLG-N010) was used as the substrate. Hydrolysis of ϵ -NAD⁺ generates fluorescent ϵ -ADP-ribose, which was monitored by measuring the increase in fluorescence at 410 nm.”

-85: it might be more clear to refer to this complex as the apo DSR2 complex or apo DSR2 tetramer throughout this manuscript, to avoid confusion with monomeric DSR2/DSR2 protomers.

Thank you. We have changed “apo DSR2” to “apo DSR2 tetramer” in the revised manuscript.

-88: clearly the domains are identifiable, they are described in the next sentence. Do the authors mean they have no known homology to other known domains?

Thank you. We have rewritten the description of the three domains as follows for clarity in the manuscript.

Lines 67-68, “Each DSR2 protomer consists of a conserved N-terminal Sirtuin (SIR2) domain, a C-terminal domain (CTD) and a middle domain (MD)”

-97: beyond just giving a reference here, the authors might provide the information that leads to this hypothesis (e.g. 'as observed for..')

Thank you. We have revised the manuscript according to this suggestion.

Lines 71-73, “The CTD adopts a horseshoe-shaped α -helical solenoid structure, which may play a role in interacting with other proteins and mediating the assembly of protein complexes, as observed for the solenoid domain of PI3K α involved in the docking of p85 α .”

-108: are conserved in both proteins

Thank you. We have revised the manuscript according to your suggestion.

Lines 90-91, “The NAD⁺-binding sites of ThsA (N113/H153) and DSR2 (N133/H171) are conserved in both proteins.”

-119: this is not analogous, as the SIR2 domains are homologs, their activity is also homologous.

Thank you. We have revised the manuscript according to your suggestion.

Line 99-101, “The similar assembly patterns of SIR2 in the ThsA and DSR2 systems suggest that SIR2 domains in bacterial anti-phage defense systems may share similar molecular mechanisms.”

-135: NADase free is not very accurate: it is a catalytic mutant lacking NADase activity.

Thank you. We have revised the manuscript according to your suggestion.

Lines 114-116, “To avoid the cytotoxicity of DSR2 NADase activation, we co-expressed a catalytic mutant DSR2^{H171A} lacking NADase activity with Tube protein, and successfully purified the DSR2^{H171A}-Tube complex”

-145: I assume this is the mutant, not the WT?

Thank you. We have revised the manuscript according to your suggestion.

Lines 124-126, “To gain a better understanding of how DSR2 recognizes the phage tail tube protein, we reconstituted the Tube-bound complex DSR2^{H171A}-Tube and determined its cryo-EM structure at a resolution of 3.58 Å.”

-151: huge is a arbitrary term, remove

Thank you. “huge” has been removed.

-197: this is not a hydrogen bond but a salt bridge?  the importance of this bond needs to be validated experimentally to support this claim.

Thank you. Following the suggestion, we have re-evaluated the interaction between E221 and K753. The distance between the two side chains is about 4.6 Å, which exceeds the typical bond length of a salt bridge (below figure). We thus have removed this claim from the revised manuscript. In addition, we introduced further mutation analysis within this interaction interface (Tube^{F204A/M206A}) to strengthen the evidence for our claim.

-199-207: these claims need mutational analysis for support

Thank you for your suggestion. In this paragraph, we have included the mutation analysis (Tube^{L50A/Y51A/I52A}) performed during the last revision process and expanded it by performing additional mutation analyses (Tube^{S31A/F32A}) to further support our claims.

The the mutational analysis are included in Lines 160-162: “Mutations at Site2 (Tube^{L50A/Y51A/I52A}) and Site3 (Tube^{S31A/F32A}), which disrupt the interaction surfaces, prevent the formation of the DSR2-Tube complex (Supplementary Fig.11).”

-247: I do not understand how this inhibitor can activate the complex, this should be further clarified and/or verified.

We apologize for the confusion caused by our previous wording. Our experimental results clearly show that the presence of DSAD1 does not activate DSR2; instead, it significantly inhibits the activation of DSR2 by Tube. The revised description is below.

Lines 212-217, “The *in vitro* NADase activity assay shows that the presence of DSAD1 significantly inhibited the activation of DSR2 by Tube (Fig.5I). The residual activity observed could be due to dissociation of DSAD1 from DSR2, allowing DSR2 and Tube to form an active complex. Alternatively, Tube proteins could bind to empty sites in the DSR2-DSAD1 complex, resulting in partial activity. ”

REVIEWERS' COMMENTS

Reviewer #2 (Remarks to the Author):

Dear editor, dear authors,

The authors have now fully addressed my remaining concerns, and the quality of their manuscript has increased significantly. The manuscript provides an important advance of our understanding of the DSR2 systems (and inhibition thereof), and experiments now well-support conclusions drawn. Furthermore, the manuscript reads very well now (I have indicated some minor typos/unclarities below). I congratulate the authors with their fantastic work.

Typos/unclarities:

-6-144: the side chains (plural)

-6-145: if these residues are important, perhaps good to name the specific residues here.

-8-199: due to steric clashes or due to a steric clash

-10-261: the striking discovery of what?

Reviewer #2

The authors have now fully addressed my remaining concerns, and the quality of their manuscript has increased significantly. The manuscript provides an important advance of our understanding of the DSR2 systems (and inhibition thereof), and experiments now well-support conclusions drawn. Furthermore, the manuscript reads very well now (I have indicated some minor typos/unclarities below). I congratulate the authors with their fantastic work.

We sincerely thank the reviewer for all the valuable suggestions, which have greatly contributed to improving the quality of our manuscript.

Typos/unclarities:

-6-144: the side chains (plural)

-6-145: if these residues are important, perhaps good to name the specific residues here.

-8-199: due to steric clashes or due to a steric clash

Thank you. We have revised the manuscript according to your suggestion.

-10-261: the striking discovery of what?

Thanks for the suggestion. We have made the following revisions to the manuscript.

-10-261: “The recent surge of interest in bacterial pattern recognition receptors has led to remarkable discoveries of various bacterial immune systems that directly recognize phage-encoded proteins and initiate cell death mechanisms as a defense strategy against phage invasion.”